# Understanding the biosynthesis of human IgM SAM-6 through a combinatorial expression of mutant subunits that affect product assembly and secretion

Haruki Hasegawa[ID][1]*, Songyu Wang[1], Eddie Kast[2], Hui-Ting Chou[3], Mehma Kaur[1], Tanakorn Janlaor[ID][1], Mina Mostafavi[1], Yi-Ling Wang[1], Peng Li[1]

1 Discovery Protein Science, Department of Large Molecule Discovery and Research Data Science, Amgen Inc., South San Francisco, CA, United States of America, 2 Molecular Analytics, Department of Biologic Therapeutic Discovery, Amgen Inc., South San Francisco, CA, United States of America, 3 Structural Biology, Department of Small Molecule Therapeutic Discovery, Amgen Inc., South San Francisco, CA, United States of America

* harukih@amgen.com

**Data Availability Statement:** All relevant data are within the manuscript and its Supporting Information files.

## Abstract

Polymeric IgMs are secreted from plasma cells abundantly despite their structural complexity and intricate multimerization steps. To gain insights into IgM's assembly mechanics that underwrite such high-level secretion, we characterized the biosynthetic process of a natural human IgM, SAM-6, using a heterologous HEK293(6E) cell platform that allowed the production of IgMs both in hexameric and pentameric forms in a controlled fashion. By creating a series of mutant subunits that differentially disrupt secretion, folding, and specific inter-chain disulfide bond formation, we assessed their effects on various aspects of IgM biosynthesis in 57 different subunit chain combinations, both in hexameric and pentameric formats. The mutations caused a spectrum of changes in steady-state subcellular subunit distribution, ER-associated inclusion body formation, intracellular subunit detergent solubility, covalent assembly, secreted IgM product quality, and secretion output. Some mutations produced differential effects on product quality depending on whether the mutation was introduced to hexameric IgM or pentameric IgM. Through this systematic combinatorial approach, we consolidate diverse overlapping knowledge on IgM biosynthesis for both hexamers and pentamers, while unexpectedly revealing that the loss of certain inter-chain disulfide bonds, including the one between µHC and λLC, is tolerated in polymeric IgM assembly and secretion. The findings highlight the differential roles of underlying non-covalent protein-protein interactions in hexamers and pentamers when orchestrating the initial subunit interactions and maintaining the polymeric IgM product integrity during ER quality control steps, secretory pathway trafficking, and secretion.

## 1. Introduction

Immunoglobulin M (IgM) is the most ancient antibody class conserved across all vertebrate species [1, 2]. IgMs are also unique among immunoglobulins in that they are produced as

**Funding:** The authors received no specific funding for this work.

**Competing interests:** The authors have declared that no competing interests exist.

**Abbreviations:** CH1, heavy chain constant domain-1; DIC, differential interference contrast; ER, endoplasmic reticulum; Fc, crystallizable fragment; Fv, variable fragment, HEK, human embryonic kidney; HC, heavy chain; IF, immunofluorescent; Ig, immunoglobulin; JC, J-chain; mAb, monoclonal antibody; LC, light chain; LLPS, liquid–liquid phase separation; MALS, multi-angle light scattering; PBS, Phosphate-buffered saline; pIgR, polymeric immunoglobulin receptor; SC, secretory component; scFv, single-chain Fv; SDS-PAGE, sodium dodecyl sulfate polyacrylamide gel electrophoresis; SEC, size-exclusion chromatography; VH, heavy chain variable region; VL, light chain variable domain.

multimers. Depending on the availability of joining-chain (J-chain, JC) subunit during an assembly reaction in the endoplasmic reticulum (ER), μ heavy chain (HC) and λ or κ light chain (LC) oligomerize into pentamers or hexamers. Therefore, a single IgM protein can possess up to ten or twelve identical antigen binding sites per molecule [3]. Binding avidity gained from such polymeric formats is crucial during the primary immune response, where the multivalency is expected to compensate for the low affinity nature of the initial antibody repertoire elicited after a new antigen challenge. In addition to such 'adaptive' or 'immune' IgMs, there is also a class of IgMs called 'innate' or 'natural' IgMs that are produced in the absence of exogenous antigen exposure [4]. These natural IgMs play critical functions in immune surveillance [4, 5]. The potential values of the IgM class for human therapeutics and diagnostic tools were re-recognized during the recent SARS-Cov-2 pandemic because the high mutation rate of surface glycoprotein and the concomitant generation of escape mutants was a significant concern for the antibody-based neutralization strategy using IgGs [6–8].

Besides the multivalency, additional attributes further distinguish IgMs from IgGs. IgMs have a relatively short serum half-life of ~4–6 days in humans [5, 9] compared to ~10–21 days for IgGs [10]. IgMs can activate complement pathways much more effectively than the IgG class and are perhaps suited for immunotherapeutic strategies via complement-dependent cytotoxicity [11, 12]. While hexameric IgMs are reportedly 20-fold more potent than pentameric IgMs in activating the complement pathway to induce cytolysis [13, 14], only the pentameric IgM can cross the epithelial barrier by transcytosis through its JC component that interacts with the polymeric immunoglobulin receptor (pIgR) [15]. During the epithelial transport, pIgR ectodomain is released by proteolysis but remains associated with the pentameric IgM to give rise to a "secretory IgM," which is released from the lumenal side of the epithelium as external secretions to serve as the first line of defense against pathogens that often favor respiratory and gastrointestinal mucosal surface as the portal of entry [16]. Randall et al. (1992) [17] showed that the activated cells produced hexameric and pentameric IgMs in different proportions depending on how B cells were stimulated (i.e., LPS or IL-5). Altering the relative abundance of hexamers or pentamers under different inflammatory conditions indicates an essential adaptive mechanism, with a trade-off between high cell lytic activity and epithelial transport. Because of such functional and physicochemical property differences, some investigators categorize pentameric IgM and hexameric IgM as two distinct subclasses [12].

IgM is also regarded as one of the most complex secretory proteins produced by the cell because of its megadalton size, multimeric nature, and intricate assembly cascade [18]. The complexity and the associated biosynthetic burden of a hexameric IgM formation can be appreciated by the need for a coordinated assembly of 12 μHCs and 12 LCs into one large hexameric IgM protein complex (~1,100 kDa) while being post-translationally modified by 60 N-linked glycans, 84 intra-domain disulfides, and 30 inter-chain disulfide bridges concurrently. In the case of a pentameric IgM (~950 kDa), one of the protomers is replaced by a JC subunit that itself has 1 N-linked glycan site and 8 cysteine (Cys) residues involved in 3 intra-molecule and 2 inter-chain disulfide bridge formation. Despite the complex assembly requirements, plasma cells in the bone marrow sustain the production of multimeric IgMs in the order of $10^8$ molecules/hour/cell (or ~25,000 molecules/second/cell) [19, 20].

Prompted by the recent pandemic, there has been a renewed interest in using recombinant IgMs and IgM-like molecules as treatment options and diagnosis tools for human diseases [1, 21]. To extend our understanding of IgMs' assembly processes and secretion requirements, we characterize the biosynthetic process of a natural human IgM called SAM-6 as our model cargo. SAM-6 was originally isolated from a stomach cancer patient as an antibody clone expected to possess anti-tumor activities [22]. Its sequence revealed that both HC and LC were encoded by unmutated germline genes [23, 24]. Although attempts to evaluate the efficacy of

SAM-6 in human clinical trials did not show objective therapeutic responses as a single agent, the IgM SAM-6 fulfilled a role as the benchmark molecule when exploring the secretory capacity of a novel human cell line [25] or a plant expression system [26]. SAM-6 also helped reveal site-dependent N-linked glycan variations within an IgM molecule [27].

To gain a better mechanistic understanding and control over the production of pentameric and hexameric IgMs, we investigate SAM-6 biosynthesis in a fully recombinant setting where all the input sequences are designed and selectively expressed in a heterologous HEK293 cell platform. In particular, the importance of individual inter-chain disulfide bonds in polymeric IgM formation and secretion is thoroughly examined by mutating μHC's conserved Cys-137, Cys-337, Cys-414, or Cys-575 residue singly or in combination with or without an additional CH1 domain deletion. Likewise, the λLC's capacity to form inter-chain disulfide is ablated by deleting the C-terminal two amino acid residues containing the penultimate Cys-213. Using the collection of mutant subunits, we assess their effects on various aspects of IgM biosynthesis in 57 different construct combinations in hexamer and pentamer settings. For each condition, steady state subcellular distribution of involved subunit chain(s) is visualized to detect changes elicited by the mutation or the presence of co-expressed mutant subunit(s). We additionally focus on the induction and prevention of Russell body-related inclusion body phenotypes and the changes in intracellular detergent solubility to elucidate their relations to IgM product quality and product secretion. In this combinatorial mutagenesis approach, we unexpectedly demonstrate that not all the prescribed inter-chain disulfide bridge is required for the biosynthesis of polymeric IgMs. In other words, the loss of inter-chain disulfide bonding at specific positions is tolerated for IgM assembly and secretion. Similarly, differential effects of mutations are observed depending on whether they are introduced to hexamers or pentamers. The finding illustrates the crucial roles of underlying non-covalent protein-protein associations that not only orchestrate the initial subunit assembly interactions but also maintain the polymeric IgM product integrity during the ER quality control steps, secretory pathway trafficking, and secretion. We suggest that the abundant production of polymeric IgMs is partly underscored by such inherently robust assembly mechanics that allow the product secretion even when not all the prescribed inter-chain disulfide bonds are formed. Our approach holistically combines the analysis of intracellular and extracellular events to illustrate the common and differential requirements for the assembly and secretion of IgM hexamers and pentamers.

## 2. Materials and methods

### 2.1. Antibodies and reagents

Affinity-purified rabbit polyclonal anti-human IgG + IgM (H+L) (cat. 309-005-107) was purchased from Jackson ImmunoResearch Laboratories. Affinity-purified rabbit polyclonal anti-human λLC (cat. A019302-2) was purchased from Dako. FITC- and Texas Red-conjugated goat polyclonal anti-human μHC (cat. 2020–02 and 2020–07), as well as FITC- and Texas Red-conjugated goat polyclonal anti-human λLC (cat. 2070–02 and 2070–07) were obtained from Southern Biotech. Rabbit anti-SEC13 (cat. AF9055100) was from R&D Systems. Rabbit polyclonal anti-calnexin (cat. C4731) and rabbit polyclonal anti-SEC16A (cat. HPA005684) were from Millipore-Sigma. Rabbit polyclonal anti-giantin (cat. PRB-114P) was from Covance. Mouse monoclonal anti-human JC (clone 3C7, aka OTI3C7) (cat. TA504168) was obtained from OriGene Technologies. FITC-conjugated mouse monoclonal anti-CD147 (clone HIM6) (cat. 555962), mouse monoclonal anti-p230 (cat. 611280), mouse monoclonal anti-calreticulin (cat. 612136), and mouse monoclonal anti-SEC31A (cat. 612350) were obtained from BD Biosciences. Rabbit polyclonal anti-BiP (cat. ab21685) was from abcam. Rabbit polyclonal anti-giantin (cat. 621352) was from BioLegend. Rabbit monoclonal anti-NPC1 (cat. MA534694)

was from Invitrogen. Mouse monoclonal anti-LAMP2 (cat. sc-18822) and mouse monoclonal anti-ERp57 (cat. sc-23886) were from Santa Cruz Biotechnology. Mouse monoclonal anti-GAPDH (clone 6C5, cat. MAB374) was from Chemicon. Unless specifically mentioned, chemicals, pharmacological agents, and biological reagents used in this study were obtained from Millipore-Sigma.

## 2.2. Expression construct cloning

The amino acid sequences of a natural human IgM SAM-6 are publicly available in the US patent publication US20110207917A1. To generate a full-length µHC sequence, we used the VH sequence "SEQ ID NO 18." Because this VH appeared to lack a portion of the FR1 region, we added a stretch of missing amino acid sequence before reconstructing a full-length µHC sequence using the constant region (UniProt: P01871). To generate a SAM-6 λLC, we selected a VL sequence "SEQ ID NO 14" and fused it to the λLC constant region (UniProt: P0DOY2). Both µHC and λLC mature domain sequences were then fused to a VK1|O12 signal sequence (V BASE website, https://www2.mrc-lmb.cam.ac.uk/vbase/) to facilitate efficient ER targeting and entry into the secretory pathway. A full-length JC sequence is reported in UniProt: P01591 (GenBank: NM_144646). All expression constructs were generated by gene synthesis and fragment assembly methods. All the recombinant genes of interest were sequence-verified and cloned into a pTT5® expression vector licensed from the National Research Council of Canada.

## 2.3. Cell culture, transient transfection, and protein production

The HEK293-EBNA1(6E) cell line was obtained from the National Research Council of Canada under research license agreements. HEK293 cells were cultured in a humidified Reach-In CO2 incubator at 37˚C, 5% CO2 using FreeStyle 293 Expression Medium (ThermoFisher). Cells were maintained in a suspension culture format using disposable shaker flasks placed on Innova 2100 platforms (Eppendorf) rotating at 130 rpm. Expression constructs were transfected into HEK293 cells using a PEI-based protocol described previously [28]. To express recombinant hexameric IgM, µHC and λLC constructs or mutant constructs were co-transfected at 1-to-1 plasmid DNA weight ratio. For the pentameric IgM expression, 3 subunit chains (µHC, λLC, and JC) were co-transfected at a plasmid DNA ratio described in the text or the corresponding figure. Difco yeastolate cell culture supplement (BD Biosciences) was added to the cell culture at 24 hr post-transfection. Cell culture media were harvested on day-7 post-transfection and used for SDS-PAGE, Western blotting, SEC, and protein purification.

## 2.4. Purification of recombinant human IgM

*Hexameric IgM.* Putative hexameric IgM was first purified from the harvested culture medium by affinity chromatography using Capto L resin (Cytiva). It is important to note that although the LC isotype of SAM-6 is λ [29, 30], perhaps due to avidity, hexameric SAM-6 IgM did bind strong enough to Capto L resin marketed to purify immunoglobulins bearing κLC. Equilibration and washing buffers were 20 mM HEPES, pH 7.4, 150 mM NaCl. The elution buffer was 10 mM acetate, pH 3.5, 150 mM NaCl. IgM-containing Capto L fractions were pooled and subjected to cation exchange chromatography using Nuvia HR-S (Bio-Rad) with an equilibrating and wash buffer (20 mM acetate, pH 5.2, 30 mM NaCl), followed by elution with a linear NaCl gradient using the following elution buffer: 20 mM acetate, pH 5.2, 1 M NaCl. Purified hexameric IgM was concentrated and dialyzed into a formulation buffer (10 mM acetate, 0.26 M sucrose, pH 5.2). The quality of purified pentameric IgM was assessed by analytical SEC using HPLC (Agilent 1200 series) on a Yarra 3 µm SEC-2000 LC 300 x 4.6 mm column

(Phenomenex) in 50 mM Tris-HCl, pH 7.0, 0.5 M arginine. _Pentameric IgM._ Putative pentameric IgM was first purified from the harvested culture medium by affinity chromatography using HiTrap Protein L (Cytiva). Like hexameric IgM, pentameric IgM did also bind to the HiTrap Protein L column marketed to capture κLC-bearing immunoglobulins [29, 30]. The bound IgM was eluted from the column using 140 mM acetic acid, pH 2.8, and the eluate was neutralized by adding 1 M Tris-HCl, pH 8.0. IgM-containing fractions were pooled and diluted in 50 mM sodium acetate, pH 5.2 and loaded onto HiTrap S HP (Cytiva). IgM was eluted from the CEX column using a linear salt gradient in 50 mM sodium acetate, pH 5.2, followed by concentration and dialysis into 50 mM sodium acetate, pH 5.2, 150 mM NaCl. The quality of purified pentameric IgM was assessed by analytical SEC using HPLC (Agilent 1200 series) on a Yarra 3 μm SEC-2000 LC 300 x 4.6 mm column (Phenomenex) in 50 mM Tris-HCl, pH 7.0, 0.5 M arginine.

## 2.5. Size exclusion chromatography coupled to multi-angle light scattering (SEC-MALS)

A TSKgel G4000SWXL column, 7.8 × 300 mm, 8-μm bead, 450-Å pore (TOSOH BIOSCIENCE) and an Agilent 1260 Infinity II HPLC system were used to determine the oligomeric state of purified IgM. The HPLC was coupled to a Wyatt miniDAWN multi-angle light scattering (MALS) detector and a Wyatt Optilab rEX refractive index detector. Protein samples at a concentration of ~0.5 mg/mL (100 μL) were loaded on the column. All experiments were conducted at room temperature at a 0.5 mL/min flow rate in PBS, pH 7.0. The MW and mass distribution of the sample were then determined using the ASTRA software version 6.1.7.17 (Wyatt Technology).

## 2.6. Immunofluorescent microscopy

Transfected cells maintained in suspension culture format were seeded onto poly-D-lysine coated glass coverslips at 48 hr post-transfection. Seeded cells were then statically cultured for 24 hr before being fixed using 0.1 M sodium phosphate buffer (pH 7.4) containing 4% paraformaldehyde for 30 min at room temperature. After washing and quenching steps in PBS containing 0.1 M glycine, fixed cells were permeabilized in PBS containing 0.4% saponin, 1% bovine serum albumin, 5% fish gelatin for 15 min, followed by incubation with desired primary antibodies in the permeabilization buffer for 60 min. After three washes in the permeabilization buffer, the cells were stained with secondary antibodies for 60 min. Coverslips were mounted to slide-glass using Vectashield mounting media (Vector Laboratories) and cured overnight at 4˚C. The slides were examined on a Nikon Eclipse 80i microscope or Eclipse Ti-E microscope with a 60× or 100× CFI Plan Apochromat oil objective lens and Chroma FITC-HYQ or Texas Red-HYQ filter. Images were acquired using a Cool SNAP HQ2 CCD camera (Photometrics) and Nikon NIS-Elements imaging software.

## 2.7. SDS-PAGE and Western blotting

On day-7 post-transfection, aliquots of suspension cell culture were withdrawn from shake flasks, and the culture media were separated from cell pellets by centrifugation at 1,000 g for 5 min. Harvested cell culture media were mixed with 2× NuPAGE lithium dodecyl sulfate (LDS) sample buffer (ThermoFisher). Cell lysate samples were prepared by lysing the cell pellets directly in 1× LDS sample buffer. 5% (v/v) beta-mercaptoethanol was included in reducing sample conditions. For non-reducing conditions, 2 mM N-ethylmaleimide was included as an alkylating agent. All samples were heated at 75˚C for 5 min. To normalize the cell lysate sample loading, whole cell lysates corresponding to 12,000–12,500 cells were analyzed per lane. A

sample volume equivalent to 5 μl of the harvested culture medium was analyzed per lane to compare the differences in volumetric titers. SDS-PAGE was performed using NuPAGE 4–12% Bis-Tris gradient gel with a compatible MES SDS buffer system (ThermoFisher). Resolved proteins were electro-transferred to a nitrocellulose membrane, blocked, and probed with primary antibodies of choice. After three washes in PBS containing 0.05% (v/v) Tween-20, the nitrocellulose membranes were probed with AlexaFluor680-conjugated secondary antibodies (ThermoFisher). After three rounds of washing, the membranes were scanned to acquire Western blotting data using the Odyssey infrared imaging system (LI-COR Biosciences).

## 2.8. Negative-stain transmission electron microscopy

Grids covered with an amorphous carbon film (01843-F, Ted Pella) were glow discharged for 15 mA and 45 sec using PELOCO easiGlow. Protein sample solution (0.1 mg/mL, 3 μL) was applied to the treated grids and stained with 0.1% uranyl formate before imaged under Talos electron microscope operated at 200 kV. In total, 597 images for the hexameric IgM sample and 406 images for the pentameric IgM sample were recorded on the K3 detector using the Latitude S software package (Gatan Inc.) at nominal magnification 22,000× corresponding to 1.84 Å/pixel. For detailed analysis, 88,125 hexameric IgM particles and 52,657 pentameric IgM particles were picked on cisTEM [31] and exported to RELION [32] for 2D classification.

## 2.9. Detergent solubility assay for intracellular IgM subunits

On day-3 post-transfection, HEK 293 cells were collected by 5 min centrifugation at 1000 g. To obtain a total cell lysate, cell pellets were lysed on ice for 15 min in 20 mM Hepes, pH 7.2, 150 mM KCl, 2 mM EDTA, 1% (v/v) Triton X-100, and cOmplete™ protease inhibitor cocktail (Roche). The "total" cell lysates were subjected to 15,000 g for 60 min to separate the "soluble" fraction from insoluble "particulate" components. The insoluble components were re-suspended in the lysis buffer and adjusted to the original sample volume. An equal volume of total lysate, soluble fraction, or reconstituted insoluble particulate component was heat treated at 75°C for 5 min in a reducing LDS sample buffer. Processed samples were analyzed by Western blotting using the designated primary antibodies.

## 3. Results

### 3.1. Biosynthesis of natural human IgM SAM-6 takes place without a major secretory bottleneck in HEK293 cells

To study the process of IgM biosynthesis, we chose a natural human IgM SAM-6 as our model protein and co-expressed the μHC and λLC subunits in HEK293 cells using a plasmid-based transfection method. To detect if there is any overt trafficking bottleneck along the secretory pathway, the subcellular distribution of μHC and λLC was visualized by immunofluorescent microscopy. At steady state, both μHC and λLC subunits localized not only in the faintly detectable hazy ER-like perinuclear structure but also in punctate structures distributed broadly in the cytoplasm (Fig 1A). To obtain insights into the identity of this punctate structure, we performed co-staining experiments using a series of representative subcellular markers of secretory and endocytic pathways. However, the IgM-filled punctate structures did not co-localize with SEC13, SEC16A, SEC31A, BiP, calnexin, giantin, p230, NPC1, or LAMP2 (see S1 Fig).

Next, we examined the intracellular behaviors of individual subunit chains. Similar to what has been shown for a collection of γHC subunits derived from various IgG antibodies [33–39],

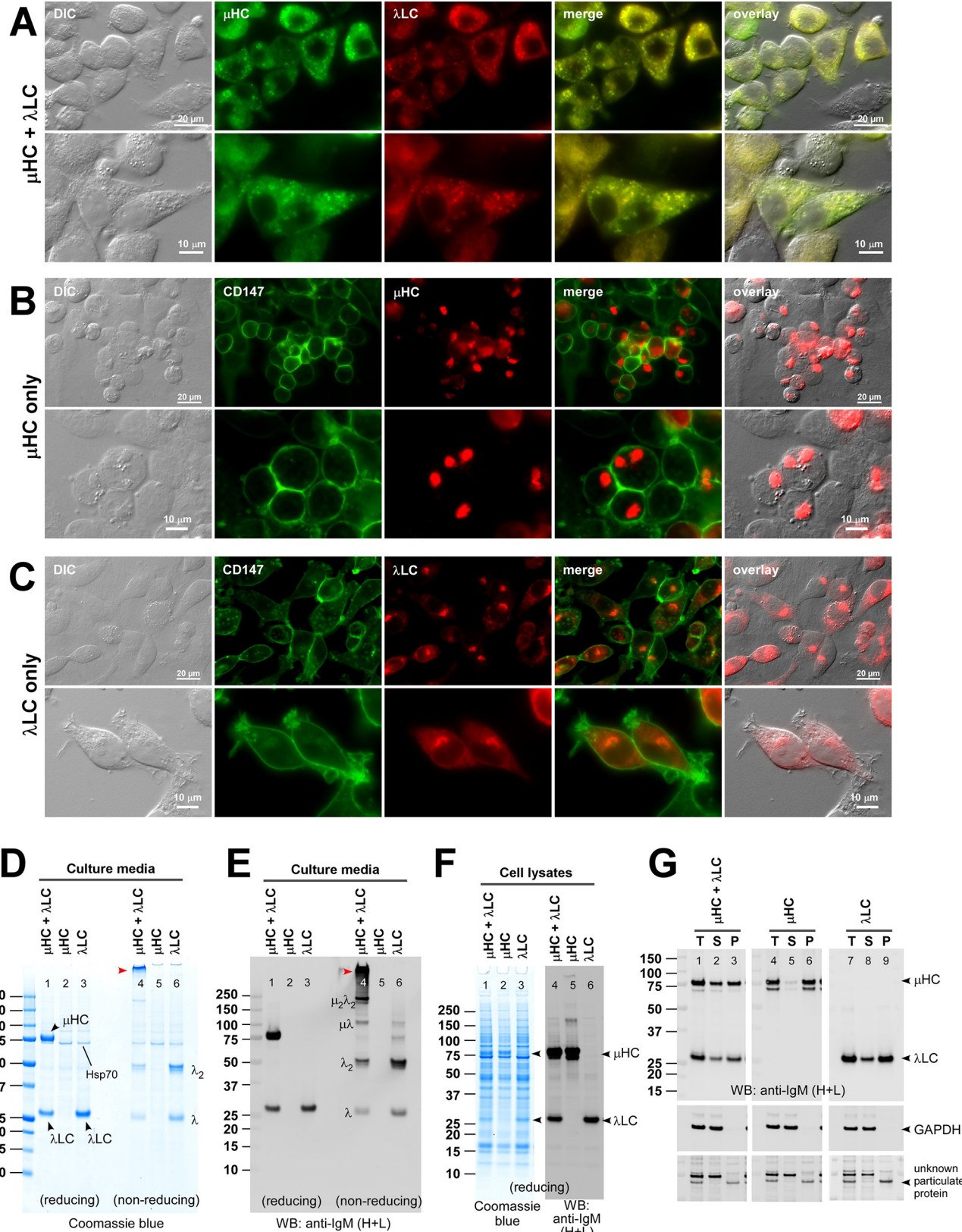

**Fig 1. Subcellular distribution and secretion of co-expressed μHC and λLC subunits during hexameric IgM expression.** Fluorescent micrographs of HEK293 cells transfected with (A) [μHC + λLC] construct pair, (B) μHC subunit alone, or (C) λLC subunit alone. (A) Co-staining was performed using FITC-labeled anti-μHC and Texas Red-labeled anti-λLC. (B) Co-stained with FITC-labeled anti-CD147 and Texas Red-labeled anti-μHC. (C) Co-stained with FITC-labeled anti-CD147 and Texas Red-labeled anti-λLC. Green and red image fields were superimposed to create 'merge' views. DIC and 'merge' were superimposed to make 'overlay' views in A. DIC and red image fields were superimposed to create 'overlay' views in B and C. (D) Coomassie blue stained gel showing the secreted IgM and subunits. HEK293 cells were transfected with [μHC + λLC] pair (lanes 1 and 4), μHC only (lanes 2 and 5), or λLC (lanes 3 and 6). Cell culture media were harvested on day-7 post-transfection and analyzed by SDS-PAGE under reducing conditions (lanes 1–3) or non-reducing conditions (lanes 4–6). The detectable subunit chain is pointed by an arrowhead and labeled in lanes 1–3. High molecular weight hexameric IgM is pointed by a red arrowhead in lane 4. Monomeric and dimeric λLC subunit is labeled in lane 6. A cell host-derived ~74 kDa protein visible in all culture media lanes was identified as human Hsp70 by mass spectrometry. (E) An identical sample set was analyzed by Western blotting. The membrane was probed with polyclonal anti-IgM (H+L) to detect μHC and λLC subunits simultaneously and any assembly intermediates composed of μHC or λLC or both. The protein band corresponding to the assembled hexameric IgM is pointed by a red arrowhead in lane 4. Identifiable assembly intermediates are labeled next to the corresponding protein bands in lane 4. (F) Cell lysate samples were prepared on day-7 post-transfection and analyzed by SDS-PAGE or Western blotting after the proteins were resolved under reducing conditions. Subunit chains are pointed by arrowhead and labeled. (G) The detergent solubility of intracellular subunits was assessed on day-3 post-transfection. Whole cell lysates were prepared under non-denaturing conditions and subjected to 15,000 g centrifugation. Total (T), soluble (S), and particulate (P) fractions were resolved in SDS-PAGE under reducing conditions. Membranes were probed with anti-IgM (H+C) (top panels) and anti-GAPDH (middle and bottom panels). Nonspecifically cross-reacting two proteins that are partitioned to soluble and particulate fractions are shown in the bottom panel.

SAM-6's μHC subunit induced globular Russell body-like globules in almost all the μHC expressing cells (Fig 1B). Co-staining experiments revealed that the globular structure co-aggregated not only with ER lumenal proteins such as BiP, ERp57, and calreticulin, but also overlapped with the staining for calnexin, an ER membrane-anchored protein, and ER-associated COPII coat proteins SEC13A and SEC31A (S2 Fig). By contrast, the globule was clearly segregated from the Golgi membranes marked by giantin and p230, or lysosome membranes shown by LAMP2 (S2 Fig). Furthermore, the intracellular μHC pool was predominantly found in the insoluble particulate fraction in a detergent solubility assay (Fig 1G, lanes 4–6). By contrast, SAM-6's λLC subunit distributed mainly to a juxtanuclear Golgi-like structure and to a lesser extent to the ER at steady state (Fig 1C) and showed no signs of gross protein aggregation. In the detergent solubility assay, unlike μHC, roughly 30%–40% of intracellular SAM-6's λLC partitioned into the soluble fraction (Fig 1G, lanes 7–9).

To study the relationship between the steady-state subcellular cargo distribution and the actual protein secretion outputs, we examined the secreted product accumulation in cell culture media on day-7 post-transfection. Co-expression of [μHC + λLC] construct pair led to an abundant IgM secretion, and both subunit chains were readily detectable in the culture media when analyzed under reducing conditions (Fig 1D and 1E, lane 1). Under non-reducing conditions, a high molecular weight protein complex corresponding to a covalently assembled putative hexameric IgM was readily detectable as a major product just beneath the loading well (Fig 1D and 1E, lane 4, red arrowhead) along with discrete assembly intermediates including λLC monomers and dimers (Fig 1D and 1E, lane 4, see labels for the assembly intermediate species). The extracellular release of assembly intermediates was described as a normal consequence of the IgM biosynthetic process [40]. Although plasma cell-specific ER resident chaperones are proposed to assist the assembly of IgMs [41–43], a non-professional secretory cell like HEK293 can also assemble and secrete this class of complex secretory cargo without major difficulties. Given that about half of μHC and λLC were found in the particulate fraction (Fig 1G, lanes 1–3), it is intriguing to postulate that specialized ER resident chaperones may enhance the secretory output by decreasing the ratio of subunits partitioned into the particulate fraction.

Extensive Russell body-like globular aggregate formation in μHC-expressing cells (Fig 1B) already alluded to a minimal free μHC secretion. In good agreement with such cell phenotype, free μHC was hardly detectable in the cell culture media (Fig 1D and 1E, lanes 2 and 5). By contrast, as implicated by its Golgi distribution at steady state (see Fig 1C), the free λLC

subunit was abundantly secreted to the culture media as a mixture of monomers and disulfide-linked dimers (Fig 1D and 1E, lanes 3 and 6). Whole cell lysate was also examined by Coomassie staining and Western blotting to show comparable expression levels of each subunit in different transfection settings (Fig 1F).

From the morphological and biochemical viewpoints, intracellular behaviors of SAM-6 μHC were very similar to those of various γHC subunits characterized in our laboratory using the same expression platform [33–39]. Namely, (1) full-length μHC had a strong propensity to induce Russell body-like detergent-insoluble globular aggregates in the absence of LC expression, (2) LC co-expression rescued the μHCs from aggregating into the inclusion body, and (3) μHC acquired a secretion competency upon pairing with the LC subunit. Although the experiments were simple, the general guiding principles of immunoglobulin assembly were recapitulated in human IgM SAM-6 using this recombinant expression platform widely used for recombinant antibody production.

## 3.2. Deleting the CH1 alone does not make the SAM-6 μHC subunit secretion competent

CH1 domain of the HC subunit is known to play critical roles in regulating the immunoglobulin biosynthesis at least by two mechanisms—the retention of free HCs by a BiP-mediated ER quality control mechanism and the covalent assembly with LC subunit via disulfide bond formation [44, 45]. Despite such importance, the CH1 domain deletion can still occur naturally via errors in gene recombination in animal models and homologous cellular systems [46]. In the case of IgGs, such CH1 deletion (denoted as ΔCH1) alone renders the mutant HCs secretion-competent by effectively bypassing the BiP-mediated ER quality control, and the mutant ΔCH1 HC dimers are secreted freely to the extracellular space in the absence of LC expression [46].

To understand the roles of the CH1 domain in μHC subunit biosynthesis and IgM assembly, we made a mutant μHC construct named μHC-ΔCH1 (Fig 2A, second row) that lacked the CH1 domain. We then assessed the steady-state subcellular distribution, product quality, and the secretion of μHC-ΔCH1 mutant by expressing itself alone or co-expressing with the λLC. Firstly, μHC-ΔCH1 expressing cells showed rounded morphology, and the μHC-ΔCH1 protein aggregated into the Russell body-like globular inclusion in those cells (Fig 2B). Cell morphology, Russell body-like inclusions, and detergent insolubility (S5A Fig, lane 7–9) were indistinguishable from those of the μHC expressing cells (see above). Likewise, co-staining with various subcellular markers revealed that the μHC-ΔCH1 protein co-aggregated with ER lumenal proteins such as BiP, ERp57, and calreticulin; and the staining also overlapped with COPII coat proteins such as SEC13 and SEC31A; as well as an ER membrane protein calnexin (S3 Fig). As before, the globular structures were separated from Golgi membranes and lysosomal membranes (S3 Fig). Secondly, in contrast to γHC's ΔCH1 counterparts, the μHC-ΔCH1 protein was secretion-competent and undetectable in the culture media by Coomassie blue staining (Fig 2D, lane 2) or by Western blotting (Fig 2E, lane 2) despite the abundant detection in cell lysates (Fig 2D and 2E, lane 6). Both the level of protein synthesis and the complete absence of protein secretion were indistinguishable between the parental μHC and μHC-ΔCH1 mutant (Fig 2D and 2E, compare lanes 1 and 2; lanes 5 and 6). Thirdly, co-expression of λLC prevented the μHC-ΔCH1 mutant from aggregating into Russell body and changed the distribution to the ER and cytoplasmic puncta (Fig 2C). Prevention of μHC-ΔCH1 Russell body formation by λLC suggested that their non-covalent interactions discouraged μHC-ΔCH1 aggregation in the ER. Interestingly, the isolated VL domain alone possessed the same ability to prevent the aggregation of μHC-ΔCH1 into Russell body (S4E

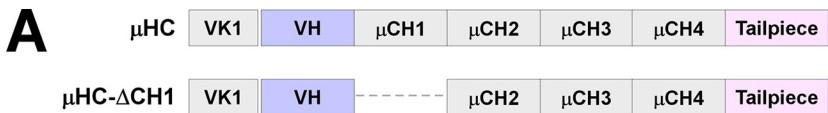

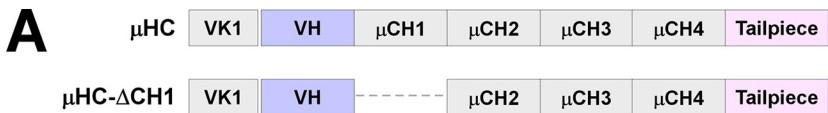

**Fig 2. Effects of CH1 domain deletion on hexameric IgM assembly and secretion.** (A) Schematic representation of SAM-6 μHC subunit and μHC-ΔCH1 mutant, which lacks the CH1 domain. Individual domain names are indicated in each box. ER targeting is driven by a heterologous signal sequence adapted from a VK1 encoding gene. (B) Fluorescent micrographs of HEK293 cells expressing μHC-ΔCH1 mutant. On day-3 post-transfection, cells were fixed, permeabilized, and immunostained with FITC-labeled anti-CD147 and Texas Red-labeled anti-μHC. (C) Fluorescent micrographs of HEK293 cells expressing [μHC-ΔCH1 + λLC] pair. Immunostaining was performed using FITC-labeled anti-μHC and Texas Red-labeled anti-λLC. (D–G) HEK293 cells were transfected with μHC (lanes 1 and 5), μHC-ΔCH1 (lanes 2 and 6), [μHC + λLC] pair (lanes 3 and 7), or [μHC-ΔCH1 + λLC] pair (lanes 4 and 8). Cell culture media were harvested on day-7 post-transfection and analyzed by SDS-PAGE under reducing conditions (D, E; lanes 1–4) or non-reducing conditions (F, G; lanes 1–4). Day-7 cell lysate samples were analyzed by SDS-PAGE (D, lanes 5–8) or Western blotting (E, lanes 5–8). Western blotting was performed using polyclonal anti-IgM (H+L) to detect μHC and λLC subunits simultaneously. A faintly detectable μHC-ΔCH1 is pointed by a black arrowhead in panel E (lane 4) and panel G (lane 4). The assembled hexameric IgM product is pointed by a red arrowhead (F, G; lane 3). Identifiable assembly intermediates are labeled next to the corresponding bands in panels F and G.

Fig), although the same VL domain could not prevent the aggregation of full-length μHC (S4D Fig). This implied that CH1, when present, needs to interact with the LC constant domain to suppress the aggregation. Fourthly, although Russell body phenotype was suppressed in [μHC-ΔCH1 + λLC] co-expression, μHC-ΔCH1 mutant remained largely secretion incompetent in that a disulfide-linked μHC-ΔCH1 dimer was only very faintly detectable by Western blotting (Fig 2E and 2G, lane 4, black arrowhead).

## 3.3. SAM-6 μHC-ΔCH1 is freed from ER retention when combined with C414/575S double mutation

Because of the presence of an 18-residue C-terminal tailpiece that drives the polymerization of IgMs, the biosynthesis of IgM is known to be more complex than that of IgGs [47, 48]. The fact that the penultimate Cys-575 residue in the tailpiece of μHC is a known substrate for a free thiol-mediated ER retention mechanism [40, 49] may explain why a mere deletion of the CH1 domain (which is a substrate for BiP-mediated ER retention mechanism) did not make the μHC subunit secretion competent (see Fig 2, and [45]).

To test whether the lack of free μHC secretion was caused by an interplay between these two ER retention mechanisms, we combined the CH1 deletion and Cys-to-Ser mutations to disable BiP-mediated and free thiol-mediated retention simultaneously. The penultimate Cys-575 in the secretory tailpiece was mutated to Ser residue with or without an additional mutation to the Cys-414 located in the CH3 domain that also plays roles in inter-chain disulfide formation between the two adjacent protomers (Fig 3A, rows 1–3) [50]. Furthermore, to study the role of these two Cys-414/Cys-575 in conjunction with BiP-mediated ER retention, the same combination of Cys-to-Ser mutations was also introduced to the μHC-ΔCH1 backbone (Fig 3A, rows 4–6).

Although C575S point mutation or C414/575S double mutation was meant to abrogate μHC's polymerization capacity, both mutant μHCs continued to show high-level Russell body-like aggregate formation that was indistinguishable from the parental μHC (Fig 3B, rows 1–3, red). Similar to the γHC-induced Russell body-like aggregates [33, 34, 36, 38, 39], those μHC globular structures co-aggregated with calnexin (Fig 3B, rows 1–3, green and merge). In good agreement with such phenotypes, these mutants were almost entirely retained in the ER and undetectable in the culture media (Fig 3C and 3D, lanes 1–3 and 7–9) despite abundant protein synthesis in the cell (Fig 3E, lanes 1–3 and 7–9).

Similarly, C575S single point mutation was not able to prevent μHC-ΔCH1 mutant from aggregating into Russell body (Fig 3B, fifth row), and the extent of Russell body formation was indistinguishable from μHC-ΔCH1 (Fig 3B, fourth row). As expected from the aggregation phenotypes, the secretion of μHC-ΔCH1 (C575S) was undetectable by Coomassie blue staining (Fig 3C and 3D, lane 5). Western blotting weakly revealed its low-level release in the form of monomers and dimers (Fig 3C and 3D, lane 11). By contrast, a combination of ΔCH1 and

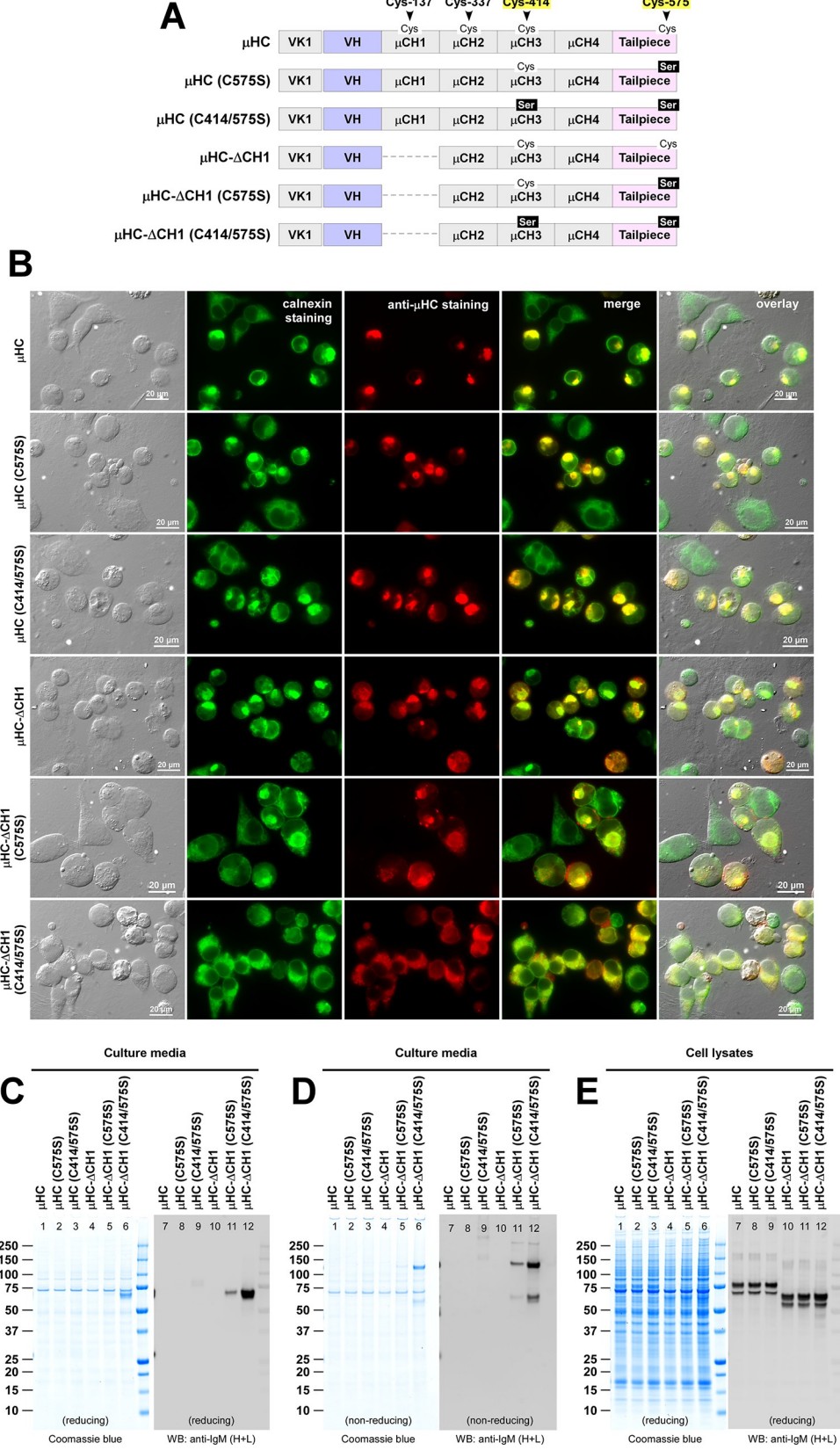

**Fig 3. Roles of Cys-414 and Cys-575 residues in μHC subunit synthesis and secretion.** (A) Schematic representation of parental SAM-6 μHC (top) and its CH1 deletion mutant μHC-ΔCH1 (fourth row) as well as their C575S and C414/575S mutant variants. The positions of key cysteine residue participating in the inter-chain disulfide bond are marked at the top. (B) Fluorescent micrographs of HEK293 cells transfected with the six constructs shown in panel A. The transfected construct name is shown on the left side of each row. On day-3 post-transfection, cells were fixed, permeabilized, and immunostained with polyclonal anti-calnexin (green) and Texas Red-labeled anti-μHC (red). Green and red image fields were superimposed in 'merge' views. DIC and 'merge' were superimposed in 'overlay' views. (C, D) Cell culture media were harvested on day-7 post-transfection and analyzed by SDS-PAGE and Western blotting after resolving the proteins under reducing (C) or non-reducing (D) conditions. (E) Day-7 cell lysate samples were analyzed by SDS-PAGE (lanes 1–6) or Western blotting (lanes 7–12). Western blotting was performed using polyclonal anti-IgM (H+L).

C414/575S not only rescued the μHC from aggregating into Russell bodies (Fig 3B, bottom row, red) but also finally liberated the mutant from the ER retention and resulted in abundant secretion as a mixture of monomers and covalent-dimers (Fig 3C and 3D, lanes 6 and 12). The simultaneous introduction of ΔCH1 and C414/575S also significantly changed the detergent solubility of the intracellular pool (see S5A Fig, lanes 7–12) and agreed well with the prevention of Rusell body-like phenotypes and acquisition of secretion competence. Despite such dramatic secretion level difference, the steady state level of synthesized proteins inside the cell was comparable (Fig 3E, lanes 4–6 and 10–12).

## 3.4. C575S and C414/575S mutations progressively block the covalent assembly of polymeric IgM

To evaluate the varying effect of Cys-to-Ser mutations on IgM biosynthesis, we co-expressed the SAM-6 λLC with each one of the SAM-6 μHC mutants illustrated in Fig 3A. Firstly, [μHC (C575S) + λLC] pair resulted in the same cytoplasmic punctate staining pattern (Fig 4A, first and second rows). The overall secretion level was also equivalent to the parental [μHC + λLC] pair (Fig 4C, lanes 1–2 and 7–8). However, the μHC (C575S) mutant could not covalently assemble the polymeric IgM species as efficiently as the parental [μHC + λLC] pair (Fig 4D, lanes 1–2). This disruptive effect was potentially attributable to the role of Cys-575 in forming a transient non-native disulfide bridge (within an IgM monomer unit) that is essential for an efficient IgM polymerization process [42]. As a result, the success rate of generating covalent polymeric IgM decreased but reciprocally increased the formation of smaller complexes such as ~200 kDa μ2λ2 protomers and ~100 kDa μλ half-molecules (Fig 4D, lanes 1–2 and 7–8). Although Cys-575 possessed essential functions to drive the covalent polymerization of mouse IgM [47, 50, 51], knocking out the Cys-575 residue alone could only partially abolish the covalent assembly of polymeric human IgM.

Secondly, the co-expression of λLC helped the μHC (C414/575S) double mutant distribute to a juxtanuclear Golgi-like structure and reticular ER-like structures (Fig 4A, third row, green) instead of aggregating into Russell body as it normally would by itself. Although the secretion of the [μHC (C414/575S) + λLC] pair was comparable to the parental [μHC + λLC] pair (Fig 4C, lanes 1 and 3; lanes 7 and 9), the secreted product quality was markedly different. When tested under a non-reducing condition, [μHC (C414/575S) + λLC] pair predominantly generated ~200 kDa μ2λ2 protomer species and could no longer secrete any covalently polymerized products (Fig 4D, lanes 3 and 9). Evidently, the [μHC (C414/575S) + λLC] pair completely lost the ability to polymerize into higher-order species covalently. Whether this mutant retains the ability to assemble a high molecular weight IgM species through non-covalent associations will be examined later by the SEC (see below, Fig 14).

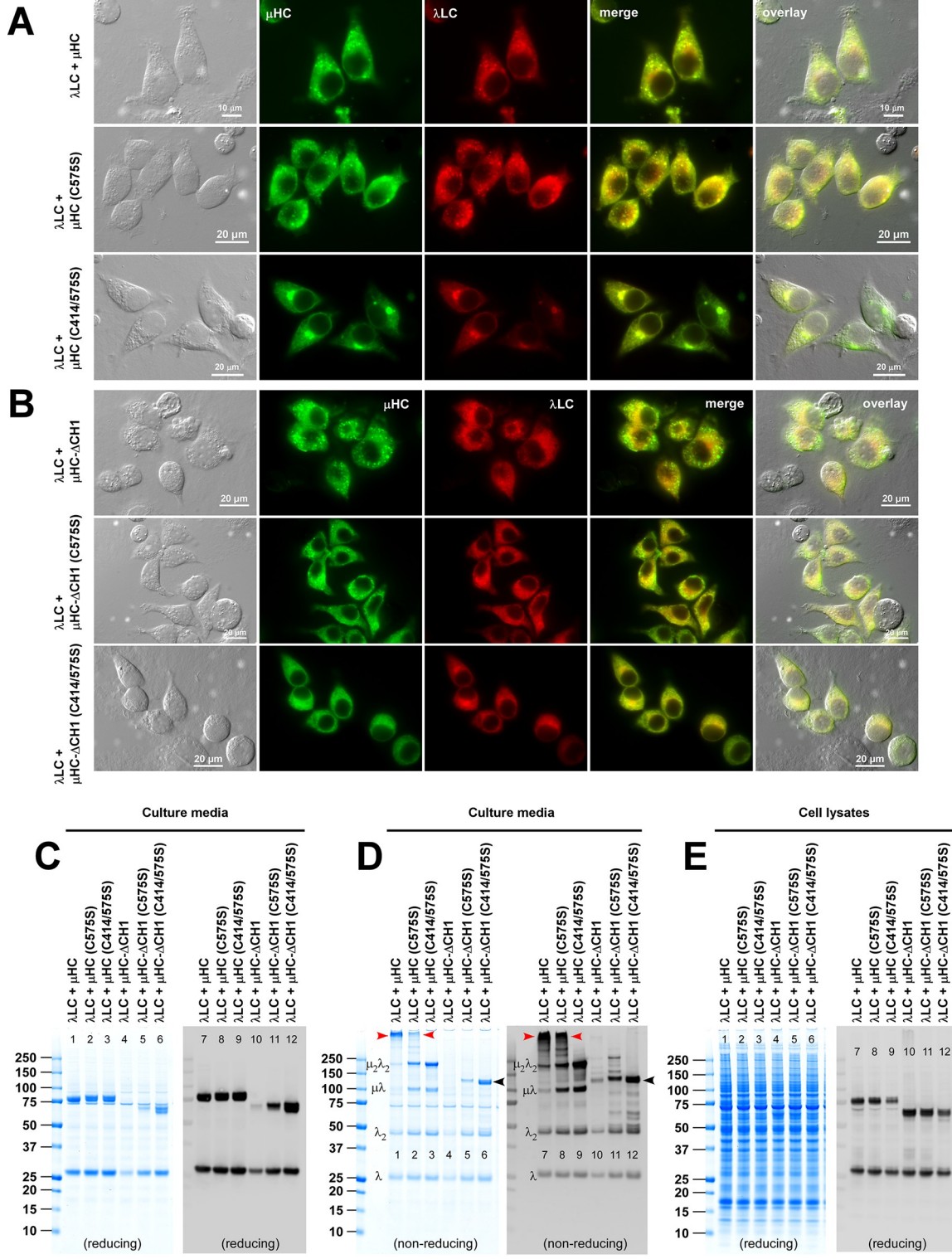

**Fig 4. Roles of Cys-414 and Cys-575 in covalent assembly and secretion of hexameric IgM.** (A) Fluorescent micrographs of HEK293 cells co-transfected with [λLC + μHC] pair (top row), [λLC + μHC (C575S)] pair (middle row), and [λLC + μHC (C414/575S)] pair (bottom row). On day-3 post-transfection, cells were fixed, permeabilized, and co-stained with FITC-labeled anti-μHC and Texas Red-labeled anti-λLC. (B) Fluorescent micrographs of HEK293 cells co-transfected with [λLC + μHC-ΔCH1] pair (top row), [λLC + μHC-ΔCH1 (C575S)] pair (second row), and [λLC + μHC-ΔCH1 (C414/575S)] pair (third row). Cell fixation, immunostaining, and image

processing were performed as described above. (C–E) The transfected construct pair is shown at the top of each lane. Cell culture media (C, D) and cell lysates (E) were harvested on day-7 post-transfection and analyzed by SDS-PAGE and by Western blotting after resolving the proteins under reducing conditions (C and E) or non-reducing conditions (D). Western blotting was performed using polyclonal anti-IgM (H+L) to detect μHC and λLC subunits simultaneously. A protein band corresponding to the covalently assembled hexameric IgM is pointed by a red arrowhead in panel D, lanes 1, 2, 7, and 8. Identifiable assembly intermediates are labeled next to the corresponding protein bands in panel D, lanes 1 and 7. Secreted μHC-ΔCH1 (C414/575S) mutant dimers are pointed by black arrowhead in panel D, lanes 6 and 12.

### 3.5. C575S and C414/575S mutation progressively increase μHC-ΔCH1 homodimer secretion regardless of λLC co-expression

Because of the missing CH1 domain, μHC-ΔCH1 series constructs cannot assemble covalently with λLC even if the λLC subunit was co-expressed. Consistent with this, μHC-ΔCH1 series mutants can only be secreted as covalently linked homodimers at best. Although protein synthesis looked comparable (see Fig 4E, lanes 4–6 and 10–12), the secretion of μHC-ΔCH1 homodimers varied significantly depending on which Cys residue was mutated (Fig 4D, lanes 4–6 and 10–12). The secretion of μHC-ΔCH1 increased when C575S point mutation was introduced (Fig 4C and 4D, lanes 4–5 and 10–11) and then more profoundly increased after C414/575S double mutations were introduced (Fig 4C and 4D, lanes 6 and 12, black arrowhead). Without the λLC co-expression, both μHC-ΔCH1 and μHC-ΔCH1 (C575S) aggregated into Russell bodies (see above, Fig 3B); however, when they were co-expressed with the λLC, the co-presence of λLC in the ER lumen effectively prevented the Russell body formation (Fig 4B, first and second rows). This illustrated the importance of non-covalent interactions between the variable regions of μHC-ΔCH1 and λLC subunit in blocking the aggregation of μHC-ΔCH1. Lastly, μHC-ΔCH1 (C414/575S) was distributed to the ER and co-localized extensively with the co-expressed λLC (Fig 4B, third row). The intracellular distribution and detergent solubility agreed well with abundant secretion level of this mutant (Fig 4C and 4D, lanes 6 and 12; S5A Fig lanes 22–25).

### 3.6. J-chain subunit is readily incorporated into polymerizing IgMs to generate pentameric IgMs

Hexameric IgM is preferentially produced when JC subunit availability is limited in the ER lumen [71, 72], while pentameric IgM predominates if sufficient JC subunit is available [17, 52, 53]. However, the amount of JC required to tip the balance from hexamer to pentamer formation has not been shown. The fate of excess free JC subunit during IgM biosynthesis is also unknown. We introduced a varying amount of JC-encoding construct into the IgM assembly reaction to determine the fine line between the hexamer and pentamer IgM formation.

The JC subunit has eight reactive cysteine residues participating in intra- and inter-molecular disulfide bond formation [54]. Because of such a cysteine-rich configuration, we first examined JC's intrinsic aggregation propensity to understand the fate of free JCs inside and outside the cells. Firstly, Western blotting analysis on cell lysates detected a ~20–23 kDa protein in the JC transfected cells (Fig 5A). Although JC is often referred to as a 15 kDa protein consisting of 137 amino acids, due to its high content of negatively charged amino acids and one N-linked glycosylation at Asn-49, the JC protein is known to run somewhat aberrantly on SDS-PAGE. Secondly, in IF imaging, overexpressed JC predominantly filled the ER-like reticular structures (Fig 5B, green) without showing co-localization with giantin, a cis-Golgi marker (Fig 5B, red). Despite the steady-state distribution to the ER, the JC did not induce notable intra-ER protein inclusion bodies. Thirdly, the ER-localizing intracellular pool of the JC subunit remained completely soluble in the detergent solubility assay (S5B Fig, lanes 1–3).

To find an optimum dose of JC subunit that can switch the IgM product quality from hexamers to pentamers, we co-transfected an increasing amount of JC construct with the [μHC + λLC] construct pair while maintaining the total amount of transfecting DNAs constant (= 10 μg) and keeping the 1-to-1 DNA ratio of μHC and λLC. In this method, the amount of DNA for the [μHC + λLC] pair decreases as the DNA for JC increases in a compensatory fashion (see Fig 5C and 5D, top of the lanes). As expected, the amount of secreted [μHC + λLC] pair in culture media (Fig 5C, lanes 1–7) and the expressed proteins in cell lysates (Fig 5C, lanes 8–14) decreased as the specific DNAs for the subunit pair decreased. Likewise, the secreted JC (Fig 5D, lanes 1–7) and the JC protein level in cell lysates (Fig 5D, lanes 8–14) increased as the relative amount of transfected JC DNA increased.

By probing the incorporation of JC into the assembling high molecular weight IgMs, we analyzed the product quality of secreted IgMs under non-reducing conditions to identify the DNA ratio that promotes pentameric IgM formation most effectively. In the range of DNA ratios tested, the lowest amount of JC DNA (2 μg) and the highest amount of [μHC + λLC] DNA pair (4 μg each) produced the greatest amount of JC-positive IgM product (Fig 5E–5G, lane 2). The switch from the hexamer to the pentamer was thus as simple as adding the JC subunit to the assembly reaction at a permissible dose range. Importantly, whenever the amount of JC exceeded the need for pentamer IgM formation, the excess free JC was released into the culture media in covalently linked polymer-like forms without being incorporated into any discrete assembly intermediates (Fig 5G, lanes 2–6). When JC was expressed by itself, secreted JCs resulted in a smear-like Western blot signal broadly distributed over a wide range of molecular weights (Fig 5G, lane 7). The fact that the smear was detected only under non-reducing conditions (compare Fig 5D and 5G, lane 7) indicated that free JCs were extensively disulfide-linked, perhaps joined like a daisy chain, using two or more of its eight reactive cysteine residues. This type of redox-sensitive smearing in Western blotting was similar to cysteine-rich cytokine IL-31 [28]. Free JC secretion came at odds with the assertion made back in 1975 that JC can only be secreted into culture media in covalent association with polymeric IgM [55]; however, when JC was incorporated into an assembling IgM, it was found only in the fully assembled IgM pentamers but not in assembly intermediates (Fig 5F and 5G).

## 3.7. IgM hexamers and pentamers are readily produced by co-expressing the required subunit chains in a permitted stoichiometry

To characterize the molecular properties of secreted polymeric IgMs, we produced and purified putative hexameric and pentameric forms of SAM-6 using the transfection conditions in Fig 5, lanes 1 and 2, respectively. Although the LC isotype of SAM-6 IgM was λ, we inadvertently found that SAM-6 IgM can be purified using Capto™ L resin (Cytiva) or HiTrap™ protein L resin (Cytiva) that are marketed to purify immunoglobulins with κLC isotype (see Materials and Methods). The purified yield was ~15 mg of putative hexamer from 500 mL of harvested culture media; whereas 3 mg of putative pentamers were purified from 300 mL cell culture.

SDS-PAGE under non-reducing conditions did not reliably discriminate the hexamers from the pentamers even when a lower % gradient gel was used (Fig 6A, a). Likewise, Western blotting using polyclonal anti-IgM (H+L) or polyclonal anti-λLC could not differentiate the hexamers from the pentamers because both forms of IgM were detected equally by these two probes (Fig 6A–6C). In Western blotting, pentamers and hexamers were distinguishable only when the purified products were probed with anti-JC (Fig 6A and 6D). Although the presence or absence of JC was revealed in the two preparations of IgM, the evidence was still missing as

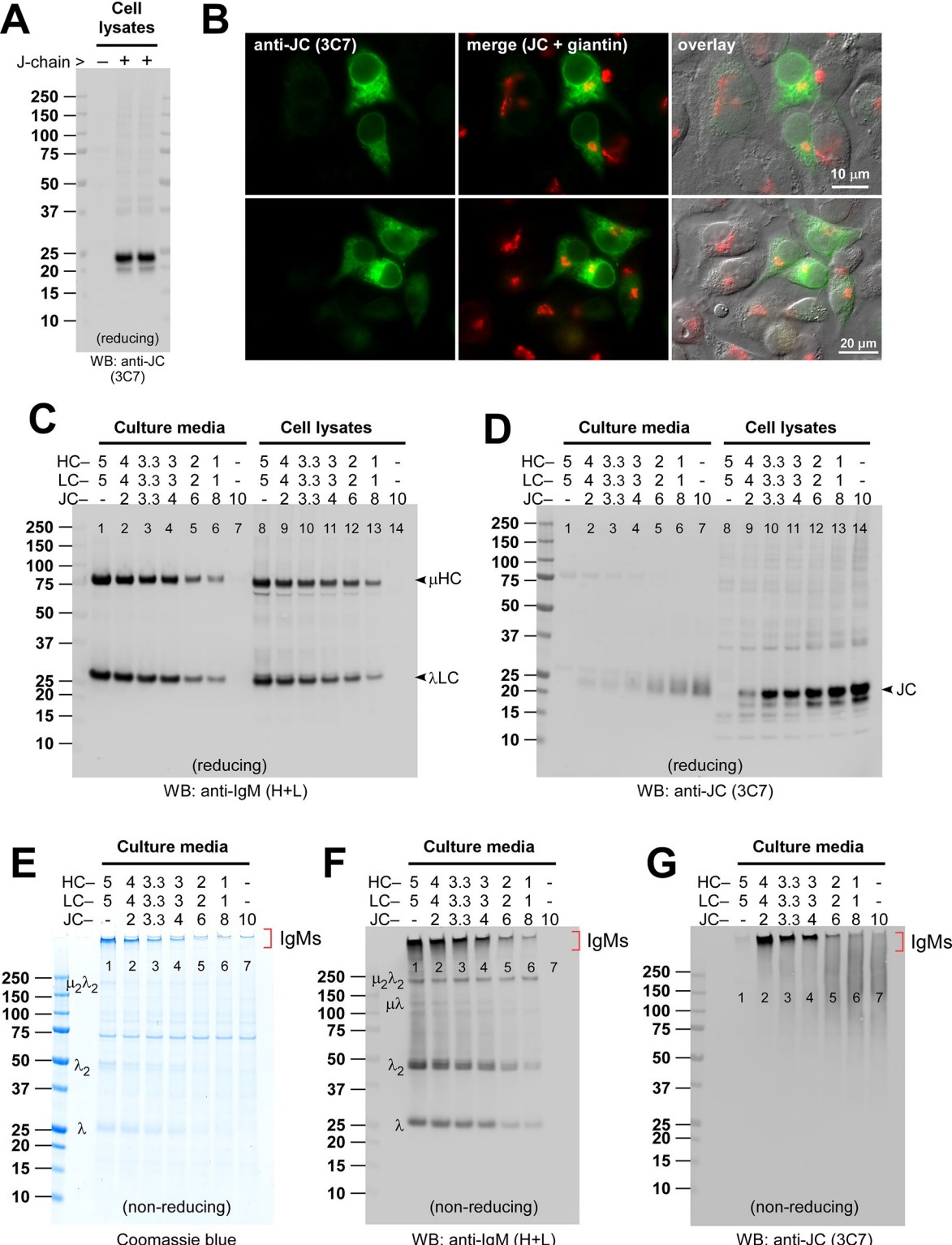

**Fig 5. Optimization of pentameric IgM secretion by titrating J-chain subunit expression.** (A, B) HEK293 cells were transfected with a human JC encoding construct, and the expression was verified by Western blotting and immunofluorescent microscopy. (A) Cell lysates were prepared on day-3 post-transfection and analyzed by Western blotting using a monoclonal anti-JC (clone 3C7). The mock-transfected cell lysate was analyzed as a control for anti-JC detection specificity. (B) On day-3 post-transfection, cells were fixed, permeabilized, and co-stained with monoclonal anti-JC (green) and anti-giantin (red). (C, D) To produce a pentameric form of IgM, three

subunit chains (μHC, λLC, and JC) were co-transfected using varying plasmid DNA ratios indicated at the top of individual lanes without changing the total amount of transfected DNA (= 10 μg). The numbers at the top of each lane represent the amount of each construct in μg. On day-7 post-transfection, cell culture media were harvested (lanes 1–7), and cell lysates were prepared (lanes 8–14) to run SDS-PAGE under reducing conditions followed by Western blotting. Membranes were probed (C) with polyclonal anti-IgM (H+L) to detect μHC and λLC subunits simultaneously or (D) with monoclonal anti-JC. (E–G) Cell culture media harvested on day-7 post-transfection were resolved by SDS-PAGE under non-reducing conditions, which were Coomassie blue stained (E) or analyzed by Western blotting (F, G). Membranes were probed with polyclonal anti-IgM (H+L) (panel F) or with monoclonal anti-JC to detect free JC and JC-containing protein complexes (panel G). In panels E and F, protein bands corresponding to λLC monomers, dimers, and assembly intermediates are labeled on the left side of lane 1. The protein band corresponding to hexameric or pentameric IgM is marked with a bracket on the right side of lane 7.

to whether the JC-positive IgMs were indeed pentamers and the JC-negative IgMs were hexamers.

In another attempt to differentiate the two types of IgMs, we examined the purified IgM products by analytical SEC. As shown in Fig 6B, both preparations showed a single main peak indicating high protein purity, but this method still did not differentiate the two types of IgMs based on the elution time or the shape of the elution peak. To distinguish the two forms of IgMs definitively, we next carried out SEC-MALS to detect the molecular mass of protein complexes present in each IgM preparation. The average mass of IgMs detected in the putative hexamer peak was 1,115 kDa ± 3.9% (see Fig 6C), and it was very close to the calculated mass of 1,135.2 kDa for a SAM-6 hexameric IgM (= 12 μHCs with 60 N-glycans + 12 λLCs). Likewise, the average mass for the putative pentameric IgM protein peak was 961 kDa ± 0.8% (see Fig 6D) and, again, it was very close to the calculated theoretical mass of 965.9 kDa for SAM-6 pentamer IgM (= 10 μHCs with 50 N-glycans + 10 λLCs + 1 JC with 1 N-glycan). From the standpoint of molecular mass, the IgMs produced without the JC consisted mainly of hexamers, while the IgMs produced with the JC were predominantly made of pentamers. To orthogonally support this conclusion, we visualized both types of IgM by negative-stain transmission electron microscopy (EM) to obtain morphological evidence on IgM polymer status. To this end, 88,125 hexamer particles and 52,657 pentamer particles were picked on cisTEM [31] and exported to RELION [32] for 2D classification analysis. The hexameric IgM was detected in the purified hexameric IgM preparation, as expected from the MALS data. Eight representative class averages are shown in Fig 6E and the images showed the hexameric arrangement of six protomers. Similarly, EM data revealed that pentamers were predominant in the JC-positive pentameric IgM preparation. Eight representative class averages obtained from the pentameric IgM preparation are shown in Fig 6F. SAM-6 IgM pentamer morphology was similar to the recently reported structure of five protomer arrangements of mouse IgM-Fcμ [56] and human IgM-Fcμ [57–59]. These pentamers have the hallmark space gap where a JC replaced a single protomer, saving space for accommodating one AIM/CD5L protein through a disulfide bond and charge-based interactions [56]. The presence of pentamer and hexamer mixtures has been known in IgM expression experiments. However, it has never been shown as clear as this study that we can qualitatively switch hexamer production to pentamer production by simply titrating transfecting DNAs for each subunit. This robust platform allowed us to evaluate the differential effects of mutations on the product quality of hexamers and pentamers separately (see below).

### 3.8. J-chains co-aggregate with μHCs via Cys-575 in the absence of LCs

In this section, we mocked situations where either μHC or λLC expression level becomes insufficient and deviates from the optimal subunit stoichiometry balance during pentameric IgM synthesis. Specifically, we examined how JC's secretory outputs and intracellular

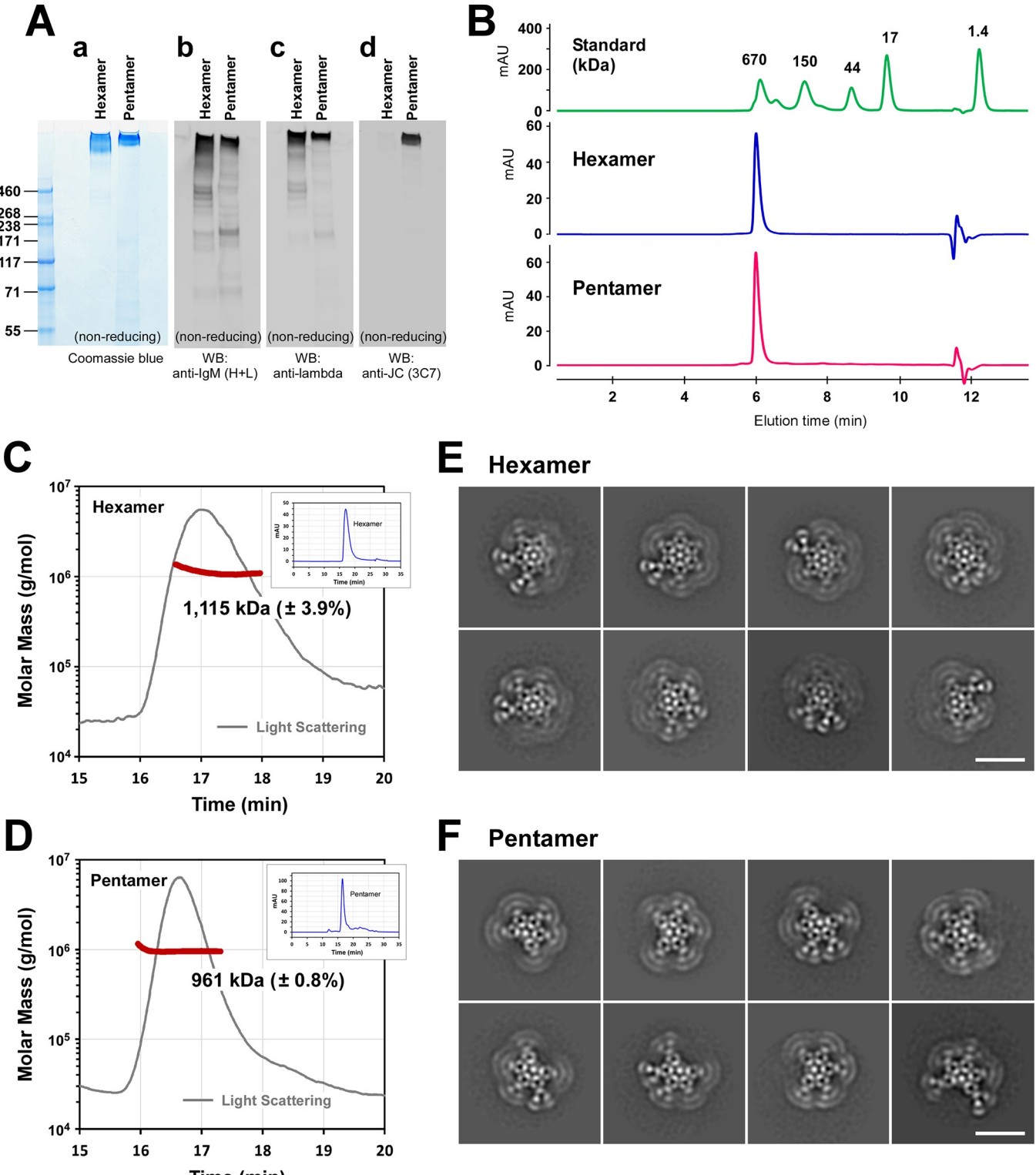

**Fig 6. Characteristics of purified recombinant hexameric and pentameric IgMs.** (A, a) Two-step purified hexameric and pentameric IgMs were resolved by SDS-PAGE under non-reducing conditions using a 3–8% Tris-Acetate gradient gel. Sample loading was 2.5 μg per lane. (A, b–d) Identically run samples were analyzed by Western blotting using three different detection probes: (A, b) polyclonal anti-IgM (H+L), (A, c) polyclonal anti-λLC, and (A, d) monoclonal anti-JC. (B) The protein purity of hexameric or pentameric IgMs was assessed by analytical SEC. About 5 μg of purified protein was injected into the column. (C, D) Purified IgM proteins (100 μg each) were analyzed by SEC-MALS. Light scattering and estimated molecular mass are plotted by solid gray line and bold brown

line, respectively. The average molecular mass for (C) hexameric IgM peak and (D) pentameric IgM peak is stated in the graphs. For calculation purposes, we arbitrarily used the mass of Man9GlcNac2 (= 1,865.64 Da) for the N-linked glycan. Simultaneously collected SEC UV trace data are shown in the inset of C and D. (E, F) Negative-stain transmission electron micrographs of purified recombinant IgM. Eight representative 2D class averages for (E) hexameric IgM and (F) pentameric IgM are shown. Scale bar, 25 nm.

behaviors were altered when λLC or μHC subunit was dropped out from the 3-chain transfection scheme.

The omission of μHC did not impact the JC distribution at all. λLC continued to distribute to the ER and Golgi (Fig 7A, green), while JC continued to localize in the ER (Fig 7A, red). Apparently, interactions between λLC and JC were insignificant, and they did not influence each other's steady-state dynamics. As a result, λLC continued to be secreted as a mixture of monomers and covalent dimers (Fig 7E, lanes 1 and 9), while the JC was secreted in a covalently chained form that showed up as a smear (Fig 7F, lanes 1 and 9). Likewise, the detergent solubility of λLC and JC remained unchanged despite the presence of each other (S5B Fig, lanes 13–15). Even if the μHC expression was completely lost during the pentamer IgM expression, such loss could go unnoticed at the cellular level because it does not result in prominent cell phenotype change.

In the λLC omission setting, μHC continued aggregating into Russell body (Fig 7B, green), while the co-expressed JC no longer distributed to reticular ER as it usually would, but instead, co-aggregated with the μHC into Russell body (Fig 7B, red and merge). In this condition, the μHC secretion was blocked as it normally would (Fig 7E, lanes 2 and 10), but the JC secretion was also prevented almost completely (Fig 7F, lanes 2 and 10). Likewise, the detergent solubility of the intracellular JC pool shifted dramatically from soluble to particulate fraction because of the co-expressed μHC (S5B Fig, lanes 1–6). These two proteins clearly interacted and co-aggregated into detergent-insoluble Russell body-like inclusions. If λLC expression was lost during the pentamer IgM production process, such an event is readily noticeable because of the dramatic change in JC's intracellular distribution and solubility.

To dissect the mechanics behind the co-aggregation of μHC and JC into Russell body, we tested C575S and C414/575S mutants. While μHC (C575S) mutant continued to aggregate into the Russell body as before (Fig 7C, green), the co-expressed JC did not accumulate into the globular structure as judged from its lack of significant co-localization with the Russell body (Fig 7C, red and merge). In this setting, although μHC (C575S) remained secretion incompetent (Fig 7E, lanes 3 and 11), the JC secretion was mostly restored (Fig 7F, lanes 3 and 11). Despite the clear change in JC's subcellular distribution and secretion, detergent solubility showed that more than half of detectable JC remained in the particulate fraction (S5B Fig, lanes 7–9).

When the μHC (C414/575S) double mutant was expressed by itself, it induced Russell bodies extensively (see above, Fig 3B, third row). Interestingly, however, co-expression of μHC (C414/575S) with JC led to inducing protein droplet-like inclusion bodies (Fig 7D). The morphology of protein droplet was similar to that of a previously characterized cryoglobulin-like scFv-Fc-stp that formed intra-ER protein droplet by liquid–liquid phase separation (LLPS) [60]. Immunostaining with anti-μHC or anti-JC antibodies could only stain and label the surface of protein droplets (Fig 7D); again, very similar to the case for scFv-Fc-stp [60]. The fact that the JC altered the intra-ER behavior of μHC (C414/575S) indicated that these two proteins interacted in the ER lumen and influenced each other's solution behavior, resulting in a notable change in inclusion body characteristics. Because both Cys-414 and Cys-575 were mutated to Ser, their interactions were most likely non-covalent. In fact, while μHC (C414/575S) remained largely non-secreting, the co-expressed JC remained secretion competent (Fig 7E,

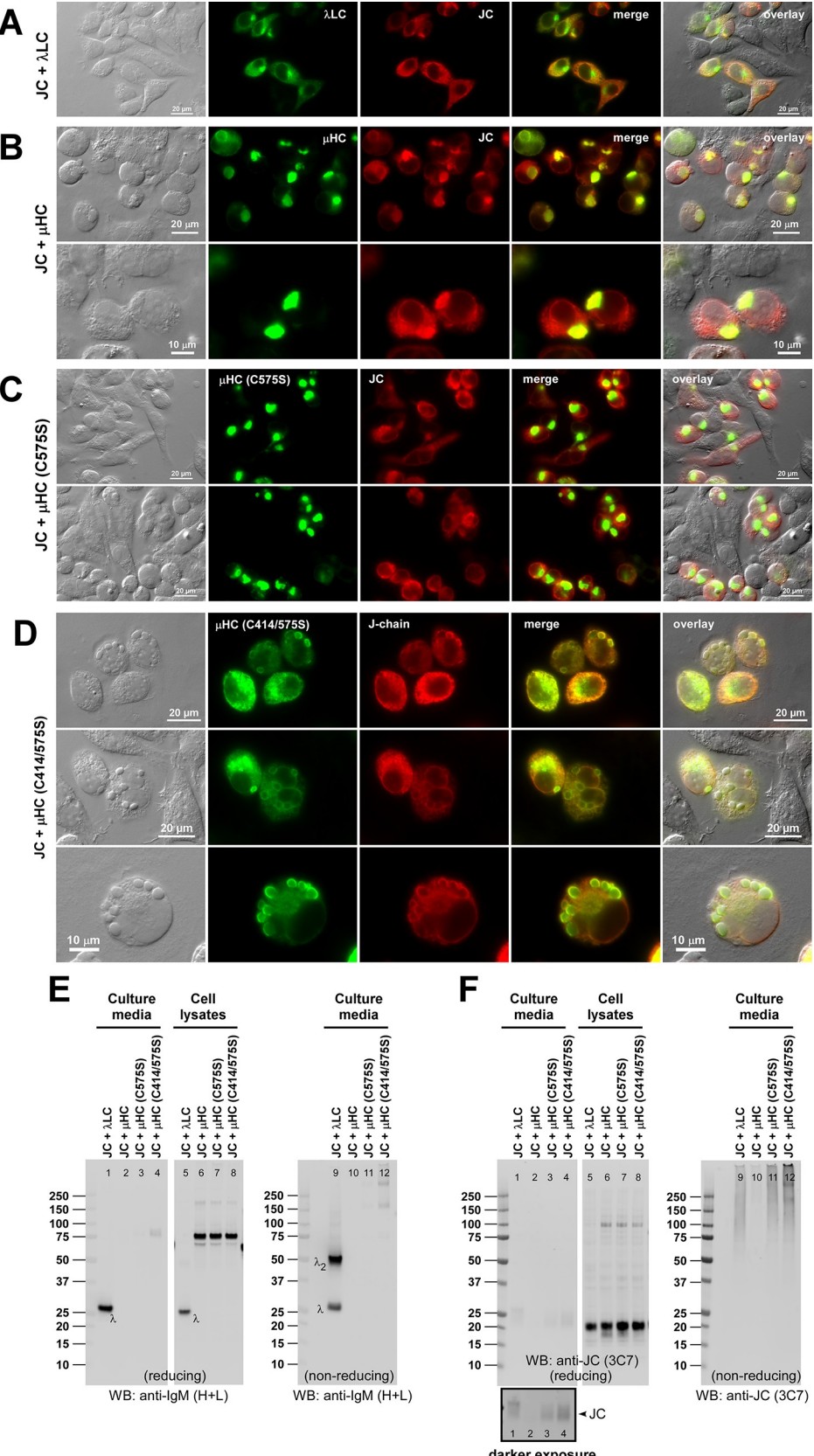

**Fig 7. Intracellular J-chain distribution is dictated by the Cys-575 of the co-expressed μHC subunit.** Fluorescent micrographs of HEK293 cells co-transfected with JC and one of the following constructs: (A) λLC, (B) μHC, (C) μHC (C575S), or (D) μHC (C414/575S). Co-transfected construct pair is also shown on the left side of each row. On day-3 post-transfection, cells were fixed, permeabilized, and co-stained with FITC-labeled anti-λLC (A) or FITC-labeled anti-μHC (B–D) and monoclonal anti-JC followed by AlexaFluor594-conjugated secondary antibody (A–D, shown in red). (E) On day-7 post-transfection, cell culture media (lanes 1–4; lanes 9–12) and cell lysates (lanes 5–8) were analyzed by Western blotting. Membranes were probed with polyclonal anti-IgM (H+L). Co-transfected construct pairs are shown at the top of each lane. (F) The same set of culture media and cell lysate samples were analyzed by Western blotting using monoclonal anti-JC. A longer exposed Western blot result is shown underneath the corresponding lanes in a black box (panel F, lanes 1–7).

lanes 4 and 12; Fig 7F, lanes 4 and 12), suggesting that their interactions are weak or transient. While JC subcellular distribution and secretion were restored to normal by the μHC (C414/575S) mutation, the JC was still partly detected in the particulate fraction in detergent solubility assay (S5B Fig, lanes 10–12).

The same set of JC co-expression experiments was performed for the μHC-ΔCH1 mutant series. The JC behavior was similarly affected by the presence of Cys-575 in μHC-ΔCH1 mutant (see S6A–S6C Fig). Likewise, the fact that μHC-ΔCH1 (C414/575S) mutant also induced the droplet-like inclusion bodies in the presence of JC demonstrated that the induction of JC-dependent droplet formation was a CH1-independent event (S6C Fig). The secretion pattern of μHC-ΔCH1, μHC-ΔCH1 (C575S), and μHC-ΔCH1 (C414/575S) in the presence of JC was identical to when these constructs were expressed by themselves (compare S6D, S6E Fig and Fig 3C, lanes 10–12).

## 3.9. Roles of CH1 domain and Cys-414/Cys-575 in pentameric IgM assembly and secretion

To appreciate the importance of the CH1 domain and Cys-414/Cys-575 residues in pentameric IgM assembly and secretion, we tested the five μHC mutants shown in Fig 3A in the 3-chain expression settings and looked for which step is affected.

The expression of all three proteins was validated by Western blotting on cell lysates, as shown in Fig 8A, lanes 1–6. The secretion outputs of individual protein components varied depending on each mutation's specific effect on folding and assembly (Fig 8A, lanes 7–12). Of note, the JC secretion was detectable when co-expressed with the parental [μHC + λLC] pair (Fig 8A, bottom panel, lane 7). This was, in fact, the only condition that allowed the productive incorporation of JC into the assembling pentameric IgMs (see Fig 8B–8D, lane 1, red arrowhead). In all other settings, the JC was expressed in the cells but not detectable in the culture media (Fig 8A, bottom panel, lanes 8–12), and the JC was not incorporated into any assembly intermediates (Fig 8D, lanes 2–6). Free JC was also expected to be secreted in lanes 8, 9, 11, and 12, but the presence of JC was only barely detectable (Fig 8A, bottom panel). Overall protein secretion was poor when μHC-ΔCH1 mutant was co-expressed with λLC and JC (Fig 8A, lane 10) because all three proteins were co-aggregated into detergent-insoluble globular Russell bodies (Fig 8F, first row; and S5B Fig, lanes 31–33).

Regarding protein quality, the JC subunit was covalently incorporated into a higher-order pentameric IgMs only when co-expressed with the parental [μHC and λLC] pair (Fig 8B–8D; lane 1, red arrowhead). JC smear was expected in lanes 2, 3, 5, and 6 (Fig 8D), but due to the low JC abundance, the blotting signal was not detectable under the employed conditions. C575S and C414/575S mutation effectively prevented the formation of covalent IgM pentamers (Fig 8B and 8C, lanes 2–3) but instead led to the secretion of ~200 kDa μ2λ2 protomers and ~100 kDa μλ half-molecules as main products (Fig 8B and 8C, lanes 2–3). By contrast, in the ΔCH1 mutant series, the λLC–μHC covalent assembly was abolished, as indicated by the

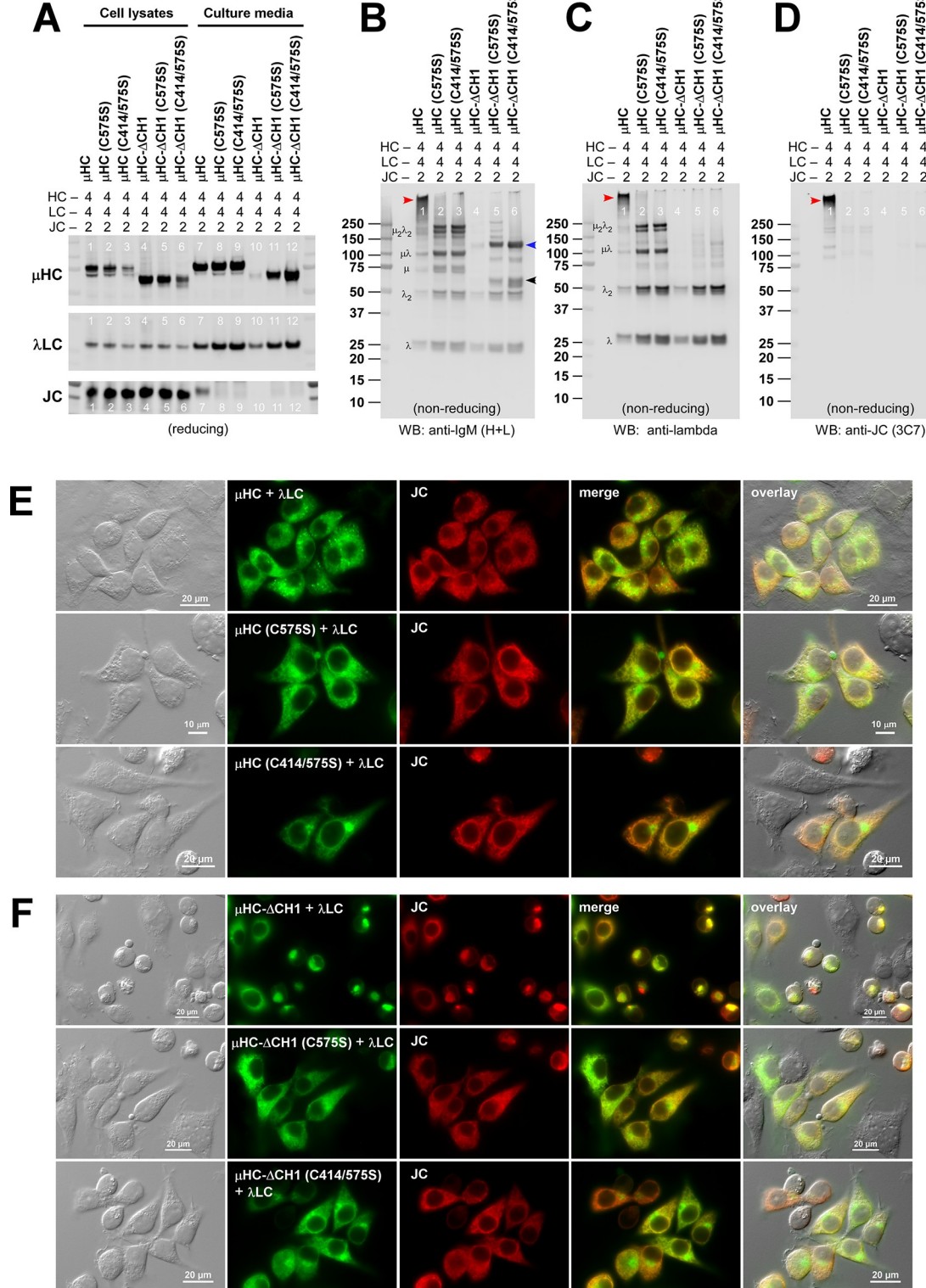

**Fig 8. Differential effects of ΔCH1, C575S, and C414/575S mutations on pentameric IgM assembly and secretion.** Three subunit chains were co-transfected to produce pentameric IgM species at the ratio of μHC: λLC: JC = 4: 4: 2. (A) On day-7 post-transfection, cell lysates (lanes 1–6) and cell culture media samples (lanes 7–12) were prepared and resolved by SDS-PAGE under reducing conditions followed by Western blotting. Membranes were probed with polyclonal anti-IgM (H+L) to detect μHC and its mutants (top panel) and the λLC subunit (middle panel) as well as with monoclonal anti-JC (bottom panel). (B–D) Day-7 cell

culture media were analyzed by Western blotting after resolving the proteins under non-reducing conditions. Membranes were probed with (B) polyclonal anti-IgM (H+L), (C) polyclonal anti-λLC, or (D) monoclonal anti-JC. Identifiable assembly intermediates are labeled next to the corresponding protein bands in panels B and C. The protein band corresponding to the assembled pentameric IgM is pointed by a red arrowhead in lane 1 of panels B–D. Secreted μHC-ΔCH1 (C414/575S) mutant monomers and dimers are pointed by black and blue arrowheads in panel B (lanes 6), respectively. (E) Fluorescent micrographs of HEK293 cells co-transfected with the following set of 3 constructs: (top row) [JC + λLC + μHC]; (middle row) [JC + λLC + μHC (C575S)]; (bottom row) [JC + λLC + μHC (C414/575S)]. On day-3 post-transfection, cells were fixed, permeabilized, and co-stained with a 1-to-1 mix of FITC-labeled anti-μHC and FITC-labeled anti-λLC to stain both subunits simultaneously (shown in green) and monoclonal anti-JC (shown in red). (F) Fluorescent micrographs of HEK293 cells expressing the following set of 3 constructs: (top row) [JC + λLC + μHC-ΔCH1]; (middle row) [JC + λLC + μHC-ΔCH1 (C575S)]; (bottom row) [JC + λLC + μHC-ΔCH1 (C414/575S)]. Transfected cells were seeded, fixed, and immunostained as above.

lack of λLC-containing protein complexes larger than the λLC dimers (Fig 8C, lanes 4–6). Although μHC-ΔCH1 mutant was non-secreting (Fig 8B, lane 4), the μHC-ΔCH1 became secretable as a mixture of monomers and homodimers when C575S or C414/575S mutation was introduced (Fig 8B, lanes 5–6; black and blue arrowheads). These mutant HC monomers and homodimers were devoid of λLC subunit (Fig 8C, lanes 4–6). Lastly, regardless of the μHC mutations, the excess free λLC was consistently secreted as a mixture of monomers and disulfide-bonded dimers under all conditions tested (Fig 8B and 8C).

In the 3-chain expression setting, both μHC and λLC continued resulting in the ER and the cytoplasmic puncta localization (Fig 8E, first row, green), while the JC remained in the ER (Fig 8E, first row, red). By contrast, if the μHC carried the C575S or C414/575S mutation, the μHC and λLC distributed to the ER and Golgi without showing the punctate staining (Fig 8E, second and third rows), while the JC remained distributed to the ER. Although μHC-ΔCH1's aggregation into Russell body was prevented by the presence of λLC (see above, Fig 4B, first row), the additional expression of JC made these subunit chains heavily aggregated into Russell body again (Fig 8F, first row). This three-chain combination resulted in the detergent-insoluble aggregate formation in agreement with the Russell body-like phenotypes (S5B Fig, lanes 31–33). Because Russell body formation was suppressed by introducing the C575S mutation to the μHC-ΔCH1 subunit (Fig 8F, second row, green) in the same 3-chain setting, it was likely that covalent linkage via Cys-575 residue was catalyzing the Russel body formation when λLC is also in the assembly mixture. However, the actual aggregation connectivity of three [μHC-ΔCH1 + λLC + JC] polypeptides is unknown. By contrast, in the 3-chain expression involving ΔCH1 (C575S) or ΔCH1 (C414/575S) subunit, both μHC mutants and λLC showed the ER and Golgi distribution (Fig 8F, second and third rows, green), while JC was consistently in the ER as expected (red). The C414/575S substitution in μHC-ΔCH1 mutant also restored the detergent solubility in 3-chain transfection (S5B Fig, lanes 34–36).

## 3.10. Covalent association between μHC and λLC is dispensable for polymeric IgM-like product formation and secretion

To characterize the role of covalent interaction between μHC and λLC in IgM biosynthesis, we disabled the λLC's capacity to form inter-chain disulfide bridge by truncating the C-terminal two amino acids (i.e., Cys-213 and Ser-214), of which the penultimate Cys-213 is essential for bridging the λLC to μHC by a disulfide bond. We designated such mutant as λLC-ΔCS (see S7A Fig). When expressed by itself, λLC-ΔCS showed a negligible ability to form disulfide-linked homodimers (see S7B, S7C Fig, lane 6) without affecting the total secretion level (S7B, S7C Fig, lanes 3 and 4) or the steady state subcellular distribution during protein synthesis (S7D, S7E Fig). By using this SAM-6 λLC-ΔCS mutant, we examined the criticality or dispensability of covalent linkage between μHC and λLC in IgM biosynthesis while keeping the μHC's CH1 domain intact.

Western blotting on cell lysates validated the protein expression of all three transfected constructs (Fig 9A, lanes 1–4). Co-transfected subunit chains were all secreted and detectable in the culture media (Fig 9A, lane 5–8, black arrowheads). Secretion levels were comparable regardless of whether full-length λLC or its ΔCS mutant was co-expressed (Fig 9A, compare lanes 5 and 6 for 2-chain transfection; lanes 7 and 8 for 3-chain transfection). Although the secretory outputs were similar, the secreted product quality somewhat differed when examined under non-reducing conditions (Fig 9B–9E). While the formation and secretion of high molecular weight IgM-like products were not impacted by λLC-ΔCS mutant both in the 2-chain and 3-chain co-expression settings (Fig 9B, compare lanes 1 and 2 for hexamer; lanes 3 and 4 for pentamer), the IgM-like products assembled with λLC-ΔCS were consistently smaller than the bona fide IgMs expressed with the full-length λLC (Fig 10B, compare lanes 1 and 2 for hexamers; lanes 3 and 4 for pentamer). In addition, the amount of secreted free λLC monomer was consistently more abundant whenever λLC-ΔCS was used (Fig 9B, compare lanes 1 and 2; lanes 3 and 4, marked by asterisks). The simplest way to interpret such product size difference and increase in detectable λLC monomers was that the association between λLC-ΔCS and μHC was labile in the presence of SDS, and the λLC-ΔCS migrated separately from the rest of the polymeric μHC backbone in SDS-PAGE.

To test if such interpretations are plausible, we performed Western blotting using the probes against different components of IgM and assembly intermediates (Fig 9C–9E). Notably, in Fig 9C, the assembly intermediates in lanes 2 and 4 were consistently ~50 kDa or ~25 kDa smaller than those species found in lanes 1 and 3; thereby indicating that two or one non-covalently associated λLC-ΔCS dissociated from the assembly intermediates during SDS-PAGE. When the same samples were probed with polyclonal anti-λLC, the IgM-like polymeric products were devoid of λLC subunit (Fig 9D, compare lanes 1 and 2; lanes 3 and 4). Although a residual amount of λLC-ΔCS was detected in lane 2, this was likely caused by an opportunistic off-pathway disulfide bonding via non-canonical cysteine connectivity [61], but a possibility of S-S re-shuffling after secretion cannot be excluded either. Lastly, the fact that the JC subunit was incorporated into the IgM-like pentamers at a comparable level as the bona fide IgM pentamers (Fig 9E, compare lanes 3 and 4) demonstrated that the covalent incorporation of JC into polymerizing μHCs occurred regardless of whether the λLC–μHC interaction is held covalently or non-covalently.

In agreement with the comparable secretion output between bona fide IgMs and "quasi" IgM-like products, the intracellular subunit distribution pattern showed the characteristic punctate pattern and was indistinguishable both in the 2-chain and 3-chain transfection settings (Fig 9F and 9G). This also suggested that non-covalent associations between λLC-ΔCS and μHC were sufficient to repress the aggregation of μHC into Russell body. In other words, non-covalent interactions between λLC-ΔCS and μHC were sufficient to hold together an IgM-like product during ER quality control steps, secretory trafficking, and even after secreted into extracellular space.

## 3.11. C137S substitution on μHC also ablates the covalent association between μHC and λLC without preventing the secretion of IgM-like products

Dispensability of inter-chain covalent interaction between μHC and λLC was unexpected because the earlier results of μHC-ΔCH1 mutant (see Fig 2) appeared to endorse the critical role of μHC–λLC covalent assembly in IgM product formation and secretion. In retrospect, deleting the entire CH1 domain might have been too invasive to retain intricate non-covalent interactions that also played roles in holding the μHC and λLC subunits together. To re-

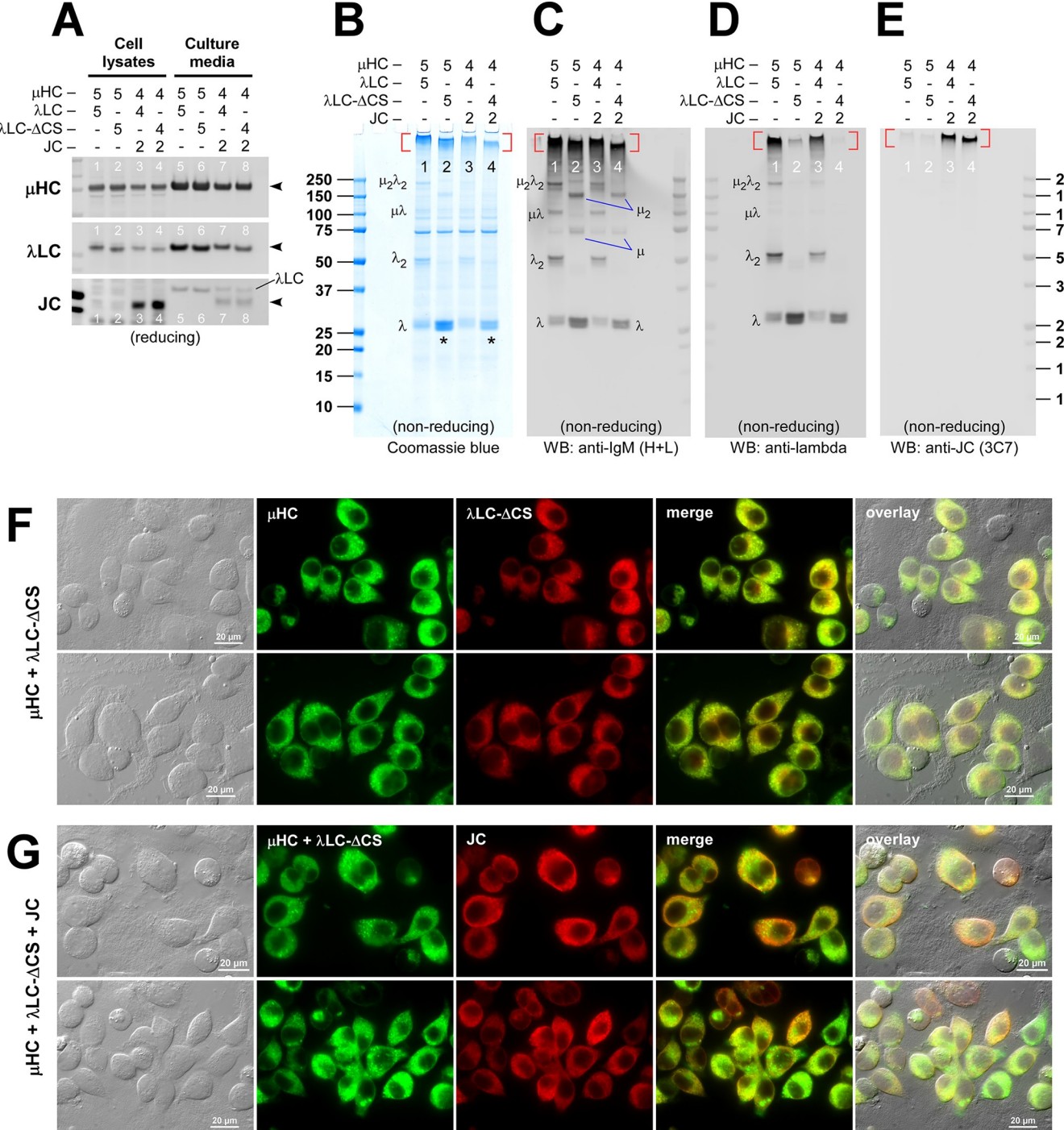

**Fig 9. Polymeric IgM-like product is assembled and secreted in the absence of inter-chain disulfide linkage between μHC and λLC.** (A) Effect of LC-ΔCS mutation on protein expression and secretion was tested in a 2-chain co-expression (lanes 1–2 and 5–6) and a 3-chain co-expression setting (lanes 3–4 and 7–8). Subunit chains were co-transfected at the DNA ratio shown at the top of each lane. On day-7 post-transfection, cell lysates (lanes 1–4) and cell culture media (lanes 5–8) were resolved by SDS-PAGE followed by Western blotting. Membranes were probed with polyclonal anti-IgM (H+L) to detect μHC (top panel) and λLC or λLC-ΔCS (second panel) or with monoclonal anti-JC (third panel). (B–E) Day-7 cell culture media were resolved by SDS-PAGE under non-reducing conditions, which were Coomassie blue stained in panel B and analyzed by Western blotting in panels C–E. Membranes were probed with (C) polyclonal anti-IgM (H+L) to simultaneously detect IgM and various assembly intermediates composed of μHC or λLC or both; (D) polyclonal anti-λLC to selectively detect IgM and subsets of assembly intermediates containing λLC or λLC-ΔCS; or (E) monoclonal anti-JC to detect pentameric IgM or any assembly intermediates containing the JC subunit. The hexameric or pentameric IgM proteins are indicated by red brackets in panels B–E. Protein bands corresponding to λLC monomers and dimers and assembly intermediates are labeled on individual gels shown in panels B–D. (F) Fluorescent micrographs of HEK293 cells co-

transfected with [µHC + λLC-ΔCS] pair. (G) Likewise, [µHC + λLC-ΔCS + JC] combination. On day-3 post-transfection, cells were fixed, permeabilized, and co-stained with (F) FITC-labeled anti-µHC and Texas Red-labeled anti-λLC or (G) a 1-to-1 mix of FITC-labeled anti-µHC and FITC-labeled anti-λLC (shown in green) and monoclonal anti-JC (shown in red).

evaluate the effect of ΔCH1 in a much less invasive fashion, we introduced a C137S mutation to µHC (see S8A Fig, second row) to block the designated disulfide bond formation with the counterpart Cys-213 of λLC without grossly affecting CH1 integrity.

The C137S mutation by itself did not alter the basic characteristics of µHC. When expressed by itself, µHC (C137S) mutant aggregated and induced Russell body-like globular inclusion to a similar degree as the parental µHC (S8D Fig). Likewise, µHC (C137S) mutant subunit was also secretion incompetent by itself (see S8B, S8C Fig). When µHC (C137S) was co-expressed with JC, µHC (C137S) and JC co-aggregated into Russell body (S9D Fig, second row), which also blocked the secretion of JC (S9A–S9C Fig, lanes 5 and 8).

To assess the effect of C137S substitution on IgM product formation, the µHC (C137S) mutant was co-expressed with the parental λLC or its λLC-ΔCS mutant in the absence or presence of JC subunit. As before, the protein expression was validated using the cell lysates to ensure that designated proteins were expressed at equivalent amounts (S10C Fig; lanes 1–4 for 2-chain; lanes 5–8 for 3-chain). The effects of µHC (C137S) mutation were unnoticeable regarding protein secretion (S10A and S10B Fig, lanes 1–4 for 2-chain; lanes 5–8 for 3-chain).

We then evaluated the secreted products under non-reducing conditions to define the effect of µHC (C137S) mutation (Fig 10A–10C). Both in 2-chain (Fig 10A, lanes 1–4) and in 3-chain (Fig 10A, lanes 5–8) expression settings, the secreted amount of high molecular weight IgM-like species was equivalent to the parental IgM even when the parental µHC was substituted with µHC (C137S) mutant. The amount of secreted IgM-like products was also comparable even when µHC (C137S) mutant was co-expressed with λLC-ΔCS mutant both in 2-chain (Fig 10A, lane 4) and 3-chain (Fig 10A, lane 8) settings. This demonstrated that IgM-like covalent product assembly and secretion took place at a similar rate regardless of whether the parental µHC and λLC were used or if µHC (C137S) and λLC-ΔCS were used in any HC–LC combinations.

Judging from the gel mobility differences, the size of hexameric IgM-like products made with the [µHC (C137S) + λLC] pair was smaller than the bona fide parental hexameric IgM (Fig 10A, compare lanes 1 and 2). Furthermore, the [µHC (C137S) + λLC] product size was indistinguishable from hexameric IgM-like products assembled by [µHC + λLC-ΔCS] pair or [µHC (C137S) + λLC-ΔCS] pair (Fig 10A, compare lanes 1–4). Similarly, the apparent gel mobility of secreted pentameric IgM-like products was clearly smaller than the bona fide pentameric IgM when µHC (C137S) or λLC-ΔCS or both chains were used to replace the corresponding parental subunits in 3-chain transfection (Fig 10A, compare lanes 5–8). These results underscored the previous section's finding that, in the presence of SDS, the association of λLC to the polymeric IgM-like backbone became labile because the point mutation abrogated the inter-chain disulfide bond formation. This conclusion was further supported by the observation that detectable free λLC monomer was consistently more abundant whenever the disulfide-forming Cys residues were ablated singly or doubly (Fig 10A, compare lane 1 to lanes 2–4). Western blotting using the probes against different constituents of the IgM further endorsed this conclusion (Fig 10B and 10C). The loss of inter-chain disulfide bond between µHC and λLC was therefore tolerated in IgM secretion.

Abolishing the inter-chain disulfide bridge formation between µHC and λLC in three different mutant combinations did not alter the steady-state subcellular distribution of participating subunits during IgM and IgM-like products biosynthesis (Fig 10D and 10E) both in

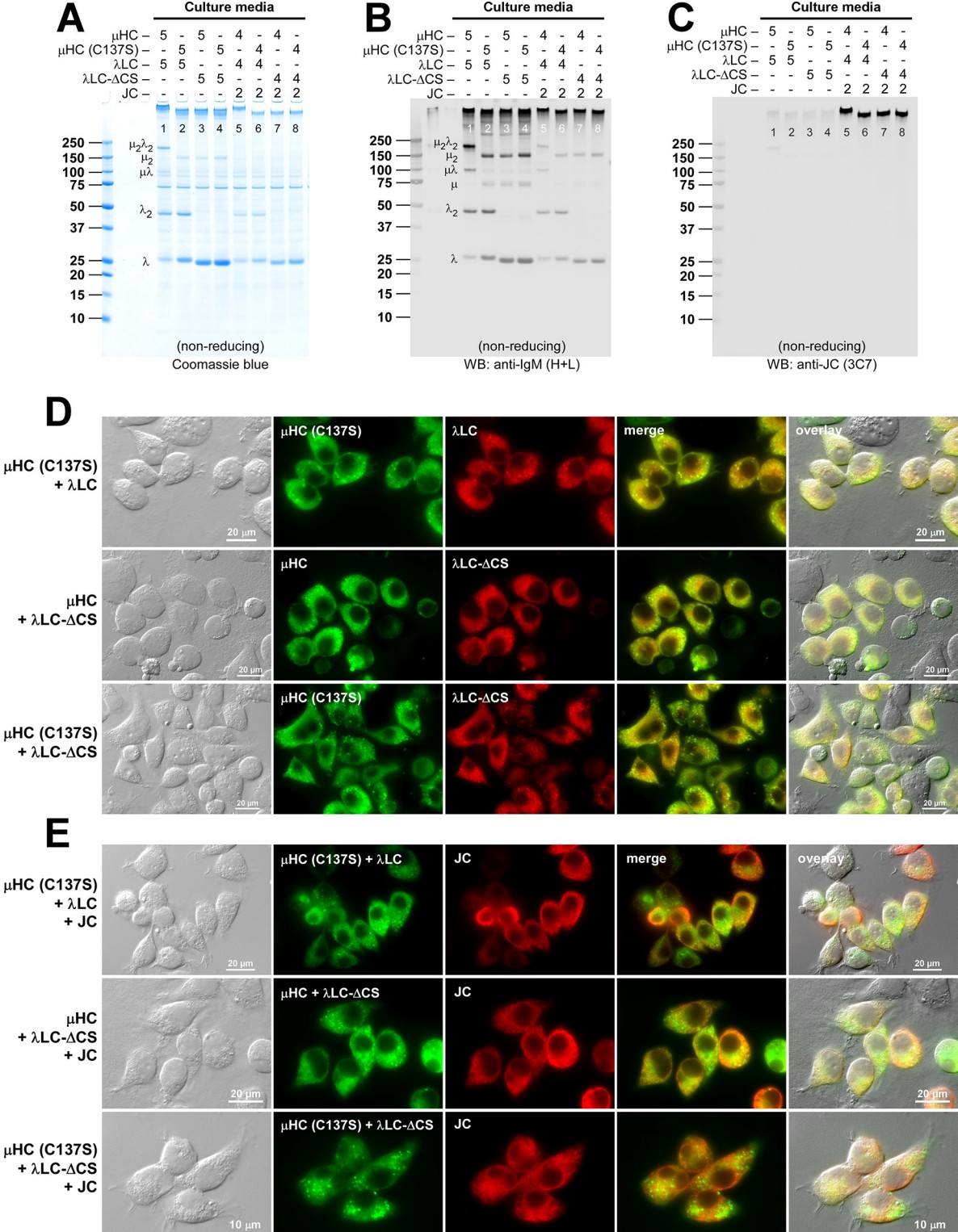

**Fig 10. Polymeric IgM-like product is assembled and secreted without covalent linkage between μHC and λLC.** (A–C) The effect of μHC (C137S) mutation on covalent polymeric IgM assembly and secretion was assessed in a 2-chain co-expression (lanes 1–4) and a 3-chain co-expression setting (lanes 5–8). Parental subunit chains and mutants were co-transfected at the DNA ratio shown at the top of each lane. Day-7 cell culture media were resolved by SDS-PAGE under non-reducing conditions, which were Coomassie blue stained (A) or analyzed by Western blotting (B, C). Membranes were probed with (B) polyclonal anti-IgM (H+L) and (C) monoclonal anti-JC. Protein

bands corresponding to λLC monomers and dimers, as well as assembly intermediates, are labeled in panels A and B. (D, E) Fluorescent micrographs of HEK293 cells co-transfected with a combination of parental or mutant subunit chains that abolish inter-chain disulfide bond between μHC and λLC. Cells were transfected with 2-chain (D) and 3-chain (E) construct sets. The transfected subunit chain combination is shown on the left side of each row. On day-3 post-transfection, cells were fixed, permeabilized, and co-stained with (D) FITC-labeled anti-μHC and Texas Red-labeled anti-λLC or (E) a 1-to-1 mix of FITC-labeled anti-μHC and FITC-labeled anti-λLC (shown in green) and monoclonal anti-JC (shown in red).

2-chain and 3-chain transfection settings. Such morphological evidence agreed well with the seemingly normal assembly and comparable secretion output of IgM-like products despite the lack of an inter-chain disulfide bridge between μHC and λLC.

## 3.12. C337S mutation abrogates the covalent assembly of polymeric IgMs but increases the complexity of secreted assembly intermediates

To further assess the roles of different inter-chain covalent associations in IgM biosynthesis, we next mutated the Cys-337 located in the CH2 domain. This Cys-337 is responsible for the inter-chain disulfide bridge that connects the pair of μHCs comprising a μHC homodimer (see S8A Fig, third row). As the position of Cys-337 would implicate, the C337S mutation is expected to split the IgM monomer unit (or protomer) into two half-molecules at the midline. However, how the C337S point mutation affects the overall IgM product quality and secretion output is not clear. The types of covalently or non-covalently associated intermediates generated from this mutation are also unknown. Expectedly, μHC (C337S) itself showed a high propensity to induce Russell body-like aggregates (S8D Fig) and was prevented from secretion (see S8B, S8C Fig, lane 3). When co-expressed with JC, μHC (C337S) mutant co-aggregated JC into the Russell body-like inclusions (S9D Fig) and suppressed the JC secretion (S9C Fig, lanes 6 and 9). Likewise, intracellular expression and secretion levels of μHC (C337S) were indistinguishable from the parental μHC in the 2-chain and 3-chain co-expression settings (S10D–S10F Fig).

Due to its critical position that splits the IgM protomer into two halves, it was not surprising that μHC (C337S) mutant subunit could no longer support the formation of covalently associated IgM-like proteins (Fig 11A, compare lanes 1 and 2 for 2-chain; lanes 4 and 5 for 3-chain). Interestingly, the C337S point mutation produced more diverse assembly intermediates than usual (Fig 11A–11D, lane 2 for 2-chain; lane 5 for 3-chain). One assembly intermediate species was larger than the typical ~200 kDa μ2λ2, but its subunit composition and stoichiometry were unclear (Fig 11A and 11B, lanes 2 and 5). When μHC (C337S) was co-expressed with λLC-ΔCS mutant, the complexity of secreted intermediates increased even more for the 2-chain co-expression setting (Fig 11A and 11B, lanes 3). Because λLC-ΔCS would dissociate from μHC (C337S) during SDS-PAGE, the apparent laddering was caused by the different numbers of μHC subunits covalently linked via the remaining Cys residues. Due to such narrow increments of size differences, it was not feasible to annotate them reliably (Fig 11B–11D). While all the tested μHC mutants failed to generate any assembly intermediates covalently associated with the JC subunit thus far, the μHC (C337S) mutant yielded two JC-positive assembly intermediates (Fig 11D, lanes 5 and 6). A similar finding on the JC-containing assembly intermediates was reported by Davis et al. [50] back in 1989. Based on their size and reactivity to different probes, these intermediates will likely be μ2λ2J and μ2J (Fig 11D, lanes 5 and 6). Whether covalent intermediates such as μ2λ2J and μ2J can assemble into loosely associated pentameric complexes is unknown. Similarly, whether the subunit stoichiometry of such "loose" complexes is identical to bona fide pentameric IgM is unclear. Attempts to analyze non-covalently associated high molecular weight species for [μHC (C337S) + λLC + JC] secretory products by analytical SEC were unsuccessful, unlike the other mutant combinations shown in Fig 14.

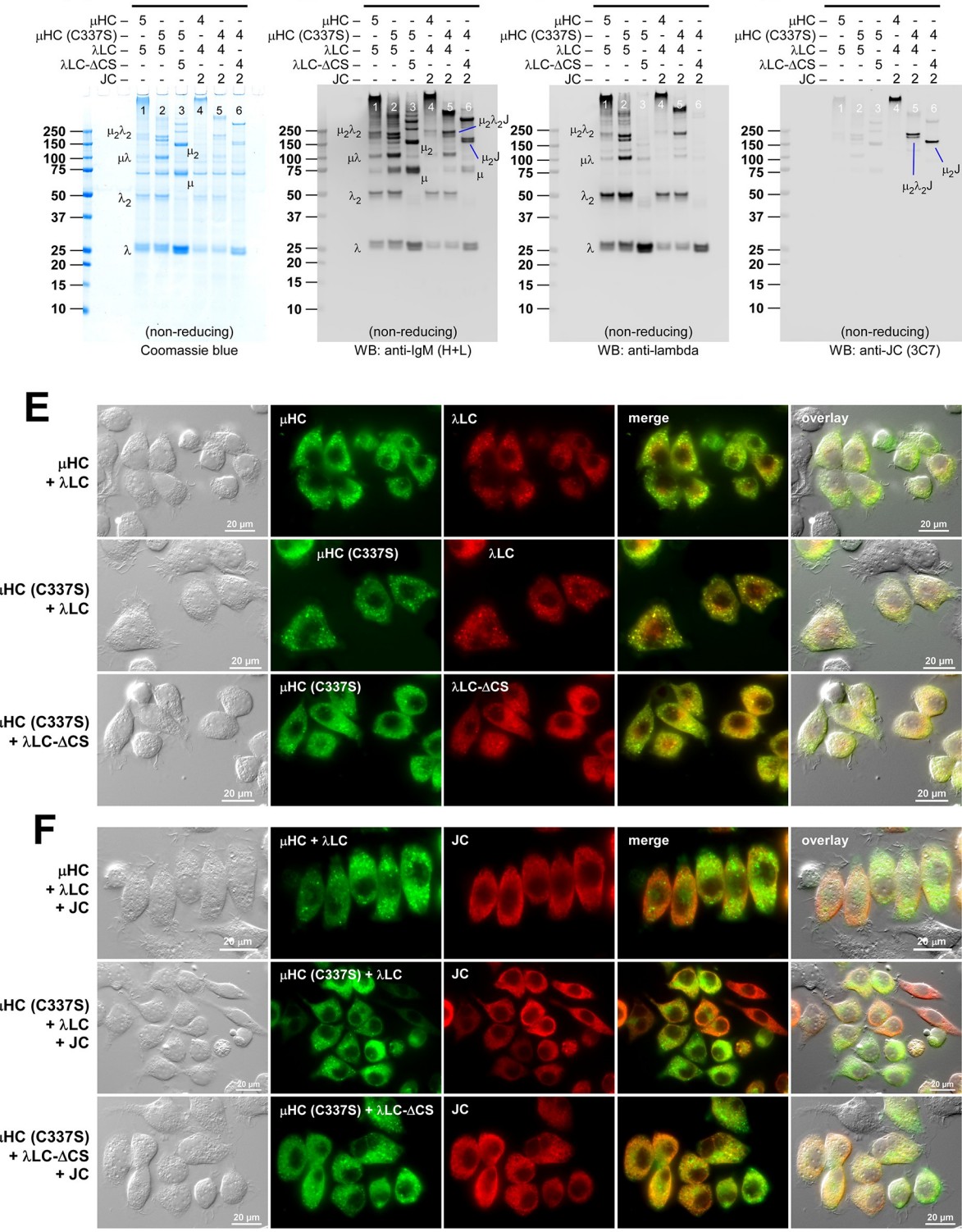

**Fig 11. Effects of C337S point mutation on hexameric and pentameric IgM product assembly and secretion.** (A–D) The role of Cys-337 in polymeric IgM formation was tested by replacing the parental µHC subunit with µHC (C337S) mutant in a 2-chain co-expression (lanes 1–3) and a 3-chain co-expression (lanes 4–6). Subunit chains were co-transfected at the DNA ratio shown at the top of each lane. Day-7 cell culture media were resolved by SDS-PAGE under non-reducing conditions, which were Coomassie blue stained in panel A or analyzed by Western blotting in panels B–D. Membranes were probed with (B) polyclonal anti-IgM (H+L), (C) polyclonal anti-λLC, or (D) monoclonal

anti-JC. Protein bands corresponding to λLC monomers and dimers, as well as other identifiable assembly intermediates, are labeled on individual gels. (E, F) Fluorescent micrographs of HEK293 cells co-transfected with a combination of parental or mutant subunit chains. Cells were transfected with 2-chain (E) and 3-chain (F) construct sets. The transfected subunit chain combination is shown on the left side of each row. On day-3 post-transfection, cells were fixed, permeabilized, and co-stained with (E) FITC-labeled anti-μHC and Texas Red-labeled anti-λLC or (F) a 1-to-1 mix of FITC-labeled anti-μHC and FITC-labeled anti-λLC (shown in green) and monoclonal anti-JC (shown in red).

When it comes to the steady state subcellular distribution of μHC (C337S), overall distribution remained the same as the parental [μHC + λLC] pair co-expression, regardless of 2-chain (Fig 11E) or 3-chain (Fig 11F) expression setting. In all cases, μHC and λLC were collectively found in the ER and cytoplasmic puncta (Fig 11E and 11F), while the JC was predominantly in the ER (Fig 11F). Although C337S mutant abrogated the production of covalently associated high molecular weight IgMs, the C337S substitution was still tolerated by the ER quality control mechanisms to allow the abundant secretion of various assembly intermediates.

### 3.13. C414S point mutation modestly affects the covalent assembly of polymeric IgMs by increasing the μ2λ2 intermediate species

Wiersma and Shulman [62] reported that "C414 is not of great importance in assembly" of mouse IgM. To test if this is also true in human IgM, we created a mutant μHC subunit with C414S substitution. The μHC (C414S) mutant subunit behaved identically to the parental μHC subunit in that it lacked secretion competency by itself (S11A–S11C Fig, lanes 2, 6, 10), formed Russell body-like globular aggregates (S11D Fig), co-aggregated with co-expressed JC (S11E Fig), and largely blocked the secretion of JC (S11A–S11C Fig, lanes 4, 8, 12). Likewise, intracellular expression and secretion levels were indistinguishable from the parental μHC in the 2-chain and 3-chain co-expression settings to produce hexamers and pentamers, respectively (S10G–S10J Fig, lanes 2).

To detect any subtle difference in the effect of μHC (C414S) mutant on IgM product quality, μHC (C414S) was compared side by side with the parental μHC, μHC (C575S), and μHC (C414/575S) both in hexamer IgM (Fig 12A–12C) and in pentamer IgM settings (Fig 12D–12G) using full-length λLC and λLC-ΔCS mutant. Firstly, μHC (C414S) mutant decreased the formation of covalent IgM hexamers (Fig 12A–12C, lanes 2 and 6) and IgM pentamers (Fig 12D–12G, lanes 2 and 6), but the protein band corresponding to μ2λ2 species increased reciprocally both in hexameric and pentameric settings (see Fig 12A and 12D, lane 2). By monitoring the secretion of μ2λ2 and μλ intermediates, C414S had the least disruptive effects compared to C575S and C414/575S mutants in both hexameric and pentameric IgM contexts. Furthermore, because μHC (C414S) mutant retains the Cys-575 residue, JC was incorporated into the pentameric IgM as efficiently as the parental IgM (Fig 12G, lanes 1–2 and 5–6), again showing the least disruptive effects on IgM assembly among the three mutant μHC subunits tested.

Similarly, as expected from the secretion levels of [μHC (C414S) + λLC] and [μHC (C414S) + λLC-ΔCS] pairs, their steady-state subcellular distribution was the characteristic combination of hazy ER-like structures and cytoplasmic puncta (Fig 12H). Even in the presence of the JC subunit in 3-chain transfection settings, the subcellular distribution pattern remained consistent, i.e., the ER-like staining and cytoplasmic puncta (Fig 12I).

### 3.14. Simultaneous ablation of all inter-chain disulfide bonds abolishes the formation of covalently assembled intermediates but does not prevent their secretion

To assess the effect of complete ablation of inter-chain disulfide bonds on polymeric IgM biosynthesis, we created a μHC (4×C>S) mutant subunit in which Cys-137, Cys-337, Cys-414,

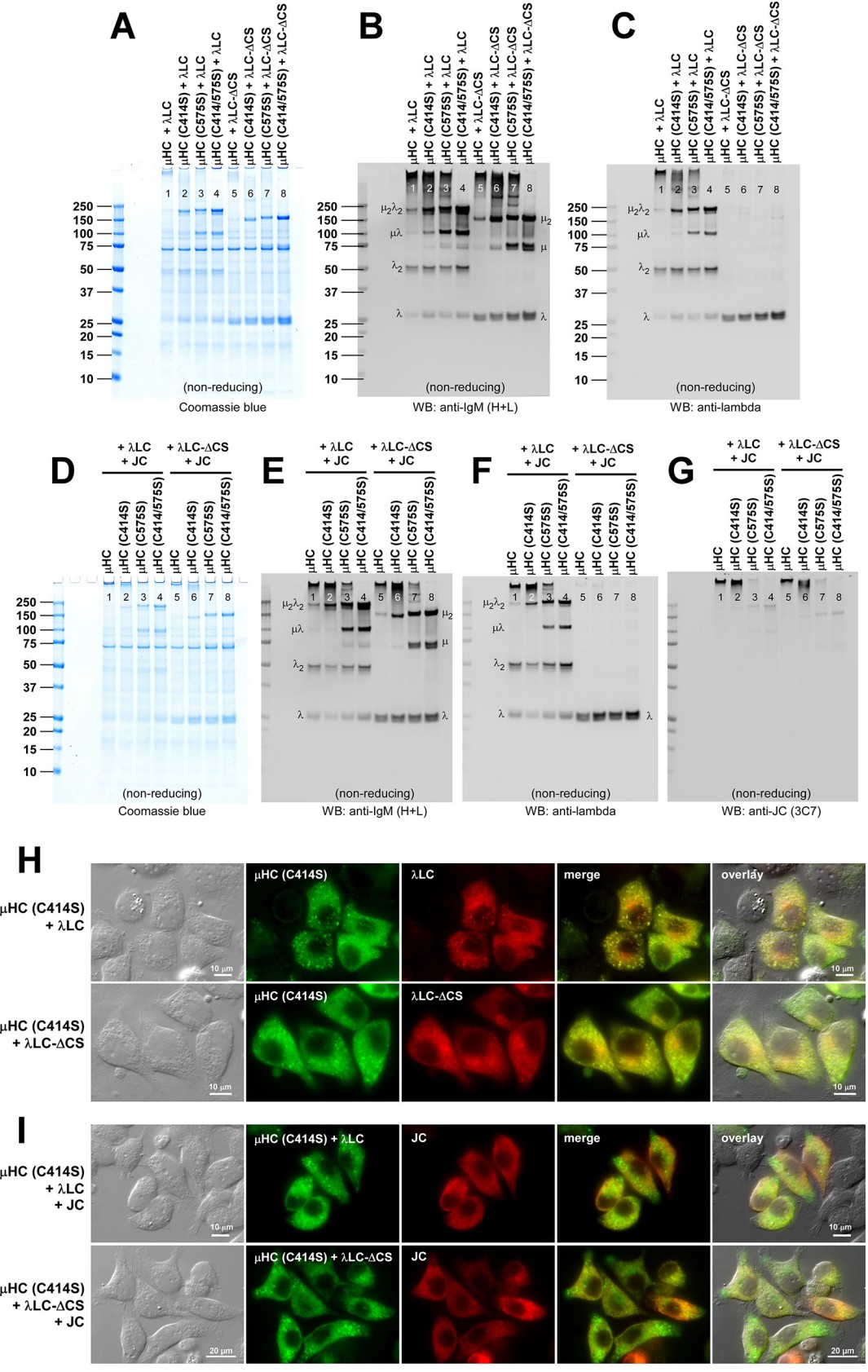

**Fig 12. Effects of C414S point mutation on hexameric and pentameric IgM product assembly and secretion.** (A–C) The role of Cys-414 in polymeric IgM formation was tested by replacing the parental μHC subunit with μHC (C414S) mutant in a 2-chain co-expression setting and compared with the effects of μHC (C575S) and μHC (C414/575S). Subunit chains were co-transfected as shown at the top of each lane. Day-7 cell culture media were resolved by SDS-PAGE under non-reducing conditions, which were Coomassie blue stained in panel A or analyzed by Western blotting in panels B and C. Membranes were probed with (B) polyclonal anti-IgM (H+L) and (C) polyclonal anti-λLC. Protein bands corresponding to λLC monomers and dimers, as well as other identifiable assembly intermediates, are labeled on individual gels. (D–G) Effects of μHC (C414S) mutation were compared with those of μHC (C575S) and μHC (C414/575S) in a 3-chain co-expression setting. As above, membranes were probed with (E) polyclonal anti-IgM (H+L), (F) polyclonal anti-λLC, and (G) monoclonal anti-JC. (H, I) Fluorescent micrographs of HEK293 cells co-transfected with a combination of parental or mutant subunit chains. Cells were transfected with a 2-chain (H) and 3-chain (I) scheme. The transfected subunit combination is shown on the left side of each row. On day-3 post-transfection, cells were fixed, permeabilized, and co-stained with (H) FITC-labeled anti-μHC and Texas Red-labeled anti-λLC or (I) a 1-to-1 mix of FITC-labeled anti-μHC and FITC-labeled anti-λLC (shown in green) and monoclonal anti-JC (shown in red).

and Cys-575 were simultaneously mutated to Ser (S12A Fig). Unlike the parental μHC that induced Russell body phenotype, μHC (4×C>S) mutant protein distributed to the ER without showing signs of aggregation (S12F Fig, second and third rows). Likewise, μHC (4×C>S) mutant was abundantly secreted by itself as monomers and, to a lesser extent, as dimers and tetramers when expressed alone (S12D, S12E Fig, lane 2). Because the μHC (4×C>S) mutant still possesses a near-intact CH1 domain (except for the C137S substitution), we expected the mutant still to be a substrate for the BiP-mediated ER retention mechanism. Yet, the free μHC (4×C>S) subunit was secreted. How this μHC (4×C>S) mutant manages to escape BiP-mediated retention without assembling with the LC is not currently understood. If μHC (4×C>S) was co-expressed with JC, both μHC (4×C>S) and JC maintained their respective ER distribution as if they were expressed individually (S12G Fig, second and third rows). As such, both proteins were secreted to the culture media as they normally would, independent of each other (S12D, S12E Fig, lanes 3–4).

To assess if μHC (4×C>S) retains any ability to assemble polymeric IgM or intermediate species covalently, μHC (4×C>S) mutant was co-expressed with the parental λLC or λLC-ΔCS mutant, both in the absence and presence of JC. As always, the protein expression level of each transfected construct was validated using the cell lysates to ensure that designated proteins were comparably expressed (S10M Fig). We also confirmed that there was no marked negative effect of μHC (4×C>S) mutation on overall protein secretion levels (S10K, S10L Fig).

To assess the effect of 4×C>S substitution on product quality, secreted proteins were resolved under non-reducing conditions (Fig 13A). Because μHC (4×C>S) can no longer form any inter-chain disulfide bridge with other proteins, including itself, the detectable products were predominantly the μHC (4×C>S) monomers without any covalent assembly intermediates (Fig 13B, lanes 3–4 and lanes 7–8). The co-expressed λLC or λLC-ΔCS was independently secreted (Fig 13C, lanes 3–4 and 7–8) without covalently associated with μHC (4×C>S). Likewise, JC did not form any covalent intermediates with μHC (4×C>S) (Fig 13D, lanes 7–8). By ablating all five cysteine residues involved in inter-chain disulfide formation (four on μHC and one on λLC), we can completely block the formation of any covalently assembled intermediates (Fig 13A and 13B, lanes 4 and 8).

Subcellular distribution of the [μHC (4×C>S) + λLC] pair and [μHC (4×C>S) + λLC-ΔCS] pair (Fig 13E, second to fourth rows) was clearly different from that of the parental [μHC + λLC] pair (Fig 13E, top row). Instead of showing the punctate staining pattern, μHC (4×C>S) distributed to the ER and Golgi (Fig 13E, second to fourth rows, green), while the co-expressed λLC or λLC-ΔCS distributed mostly to the Golgi as if these subunits were expressed independently (Fig 13E, second to fourth rows, red). Likewise, when JC was co-expressed in the 3-chain expression setting, μHC (4×C>S) and λLC or λLC-ΔCS remained distributed to the

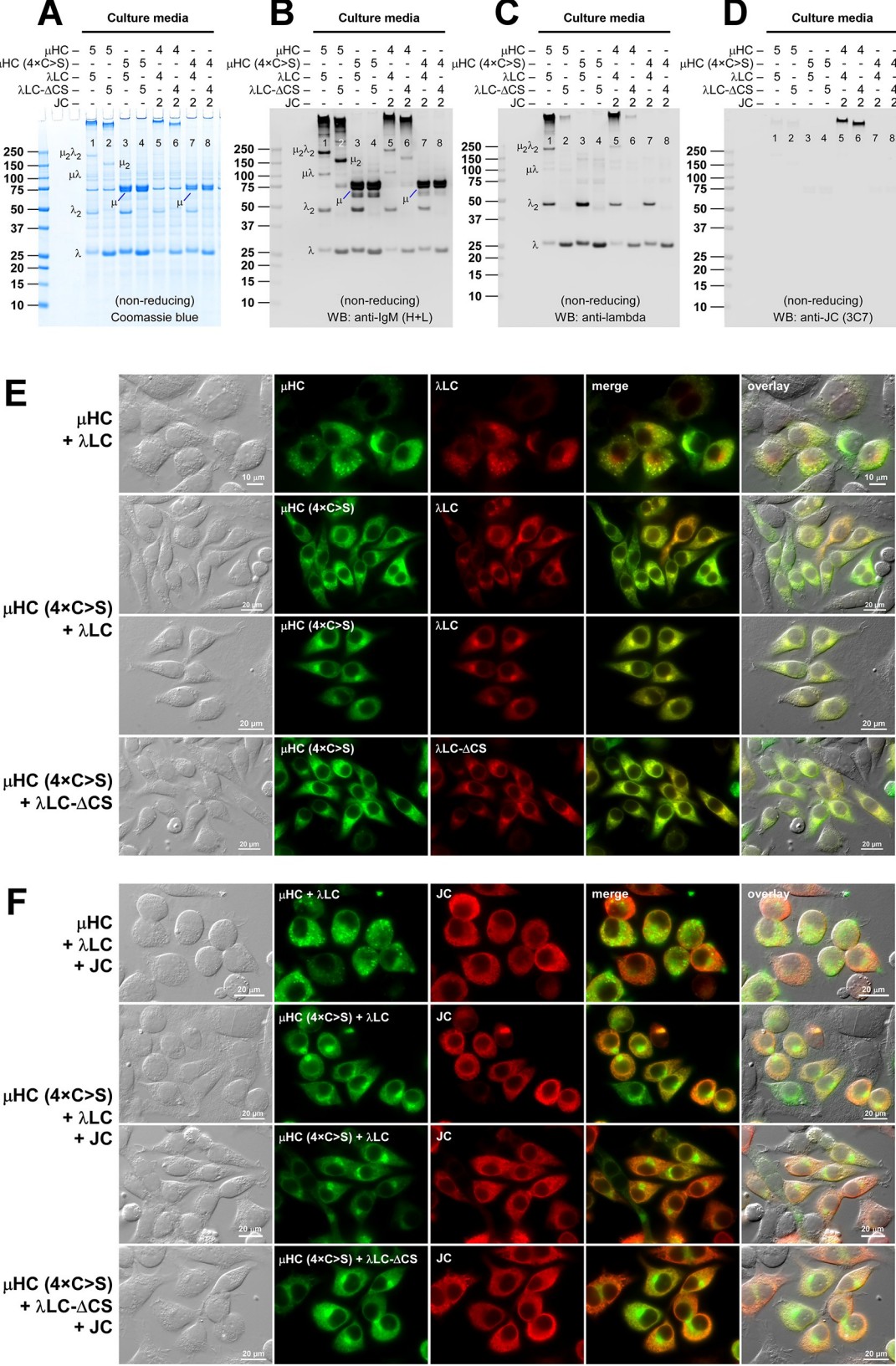

**Fig 13. μHC subunit cannot produce covalent intermediates when all four cysteine residues are mutated.** (A–D) The effect of simultaneous inter-chain disulfide bridge ablation was tested in a 2-chain co-expression (lanes 1–4) and a 3-chain co-expression (lanes 5–8) using μHC (4×C>S) mutant subunit. Day-7 cell culture media were resolved by SDS-PAGE under non-reducing conditions, which were Coomassie blue stained in panel A and analyzed by Western blotting in panels B–D. Membranes were probed with (B) polyclonal anti-IgM (H+L), (C) polyclonal anti-λLC, or (D) monoclonal anti-JC. Protein bands corresponding to λLC monomers and dimers, as well as identifiable assembly intermediates, are labeled on individual gels. (E, F) Cells were transfected with (E) 2-chain and (F) 3-chain construct sets. The transfected subunit combination is shown on the left side of each row. Cells were fixed, permeabilized, and co-stained with (E) FITC-labeled anti-μHC and Texas Red-labeled anti-λLC or (F) a 1-to-1 mix of FITC-labeled anti-μHC and FITC-labeled anti-λLC (shown in green) and monoclonal anti-JC (shown in red).

ER and Golgi, whereas JC continued to show ER localization (Fig 13F, second to fourth rows, red). Furthermore, among the co-expression conditions tested, [μHC (4×C>S) + λLC] was the only pair where both μHC and λLC were fully partitioned to the soluble fraction in the detergent solubility assay (S5A Fig, lanes 16–18).

## 3.15. Polymeric IgM product integrity is partly maintained through non-covalent associations even when certain inter-chain disulfide bonds are eliminated

A series of studies published in the 1970s [63–66] collectively suggested that polymeric IgM molecules can be reconstituted through non-covalent forces alone from a mixture of partially reduced and alkylated IgM monomers and/or H-L half-mers using various sources of IgMs and different assay conditions. Although these studies highlighted the importance of non-covalent protein-protein interactions when reconstituting polymeric IgMs from a different mixture of intermediates in test tubes, it is unknown if polymeric IgMs can assemble biosynthetically with a fewer subset of inter-chain disulfide bonds or without the inter-chain disulfide bonds altogether. To explore IgM's intrinsic capacity to assemble into polymeric formats during biosynthesis, we assessed which disulfide elimination is tolerated when generating hexameric and pentameric IgMs partly or solely through non-covalent associations.

Up to this point, the product quality of secreted IgMs, IgM-like molecules, and assembly intermediates were analyzed by SDS-PAGE under non-reducing conditions. Although this method preserves disulfide-mediated covalent associations, non-covalent interactions are destroyed because of SDS. To assess which specific inter-chain disulfide bond formation is dispensable in the formation and maintenance of high molecular weight IgM species, we examined the quality of secreted products by analytical SEC. Because most non-covalent protein-protein associations are preserved, this assay approach should give us more accurate accounts of the secreted mutant product quality under physiological (non-denaturing) settings.

To assess the effects in hexameric IgM context, the parental μHC and its six different mutant variants (i.e., C137S, C337S, C414S, C575S, C414/575S, and 4×C>S) were co-expressed with λLC (Fig 14A, panels 1–7) or λLC-ΔCS (Fig 14A, panels 8–14) in the 2-chain transfection settings. The harvested cell culture media were directly examined by analytical SEC without any purification step. The parental [μHC + λLC] pair yielded hexameric IgM and monomeric IgM as the two recognizable main products in addition to the LC dimer, LC monomer, and one unknown intermediate that eluted at ~7 min 45 sec (panel 1). Similarly, the [μHC + λLC-ΔCS] pair secreted hexameric IgM, monomeric IgM, and the LC monomer (panel 8). Likewise, μHC (C137S) mutant retained the ability to assemble hexameric IgM species through non-covalent interactions between HC and LC when co-expressed with the parental λLC (panel 2) or λLC-ΔCS (panel 9). These first four SEC results solidified the earlier results that the loss of inter-chain disulfide bond between μHC and λLC was tolerated in the hexameric IgM product assembly and secretion.

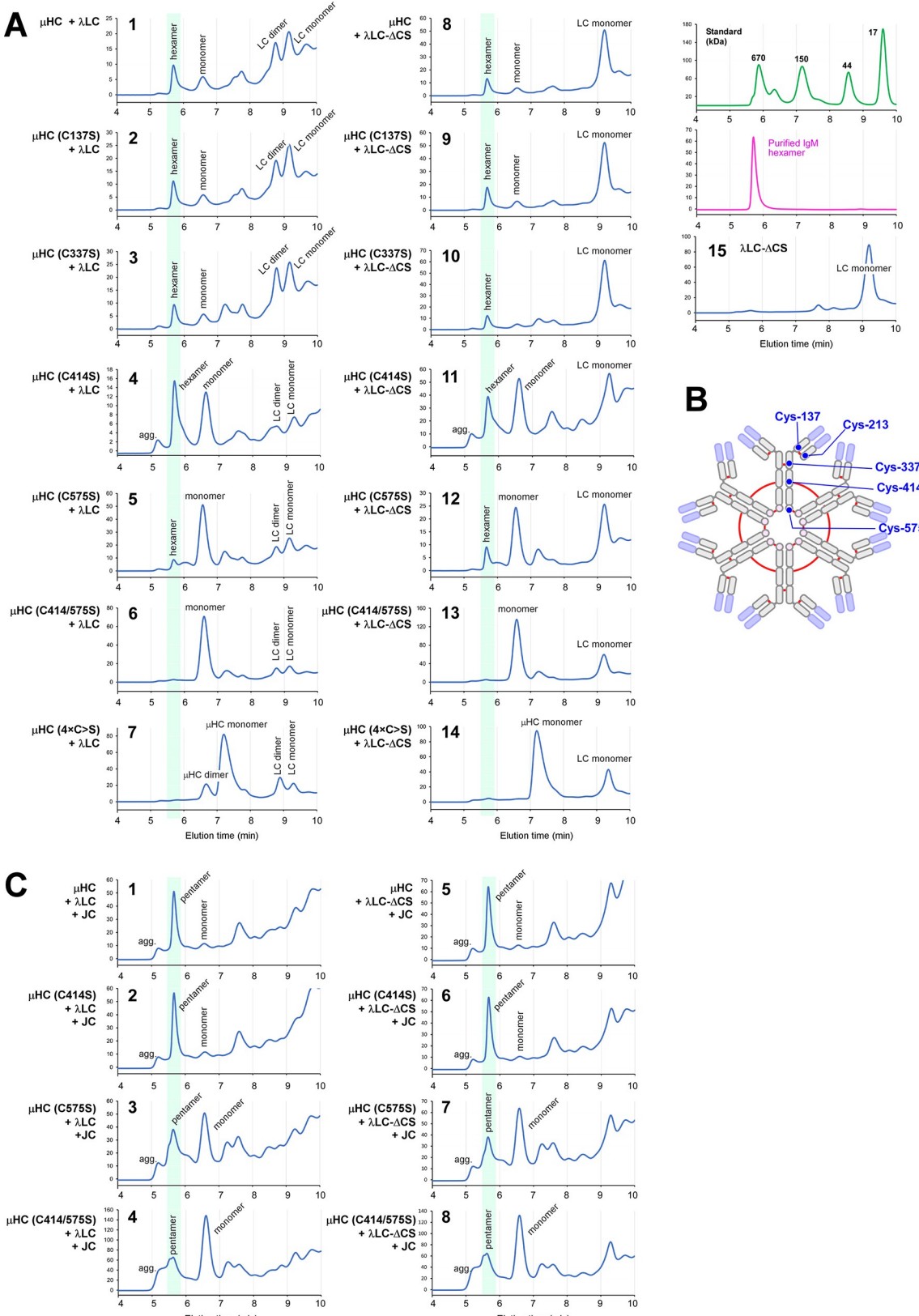

**Fig 14. The integrity of polymeric IgMs is maintained partially through non-covalent protein-protein interactions.** (A) The secreted product quality was assessed by analytical SEC under physiological conditions. The harvested day-7 cell culture media obtained from 14 different [μHC + λLC] subunit chain combinations were directly analyzed by analytical SEC. The transfected subunit chains are shown on the left side of each chromatogram. Purified hexameric IgM was analyzed as a reference for polymeric IgM elution (right-most column, second panel). In panel 15, the elution profile of λLC-ΔCS mutant is shown as a monomeric λLC protein. The elution peak corresponding to hexameric IgM is shaded in light green in individual chromatograms. (B) The position of Cys residues involved in inter-chain disulfide bonds is shown in the context of hexameric IgM structure. Solid red lines represent the inter-chain disulfide bond connectivity. (C) SEC elution chromatograms for day-7 culture media containing the parental pentameric IgM (panel 1) and its mutant series (panels 2–8). The transfected subunit chain combination is also shown on the left side of each chromatogram. The elution peak corresponding to the designated pentameric IgM is shaded in light green in individual chromatograms. An aggregated protein peak fraction eluted immediately after 5 min is indicated as "agg.".

The C337S mutation hampered the covalent assembly of polymeric IgMs (see Fig 11A, lanes 2 and 3). However, SEC analysis showed that C337S mutant IgM retained the capacity to assemble hexameric IgM partly held through non-covalent associations (panels 3 and 10). The loss of Cys-337 mediated inter-chain disulfide bond that holds two μHC together was tolerated.

Although inefficient, both μHC (C414S) and μHC (C575S) mutants weakly retained the capacity to produce covalently assembled hexameric IgM products (see above, Fig 4D, Fig 12A–12C). SEC also showed that hexameric IgM species are maintained in both μHC (C414S) and μHC (C575S) mutants (panels 4 and 5; panels 11 and 12). Comparing their effects on the hexamer to monomer ratio, the μHC (C414S) showed weaker disrupting effects by maintaining a higher hexamer ratio (compare panels 4 and 5; panels 11 and 12).

The μHC (C414/575S) double mutant was no longer able to produce and maintain hexameric IgM species neither through covalent nor non-covalent interactions (panels 6 and 13). Instead, the monomer unit of IgM predominated the secretory products. Similarly, when all four cysteine residues involved in the inter-chain disulfide formation were mutated to serine, μHC (4×C>S) mutant was almost exclusively secreted as μHC (4×C>S) monomers and lost the ability to form any assembly intermediates through non-covalent forces (panels 7 and 14). Furthermore, SEC analysis on λLC-ΔCS mutant by itself also suggested that non-covalent associations alone could not produce the LC dimers, indicating an obligatory role of the disulfide bridge formation in λLC dimer formation (panel 15).

A similar set of SEC assays was also performed for a subset of pentameric IgM mutants using harvested cell culture media (Fig 14C). As in hexameric IgM, the covalent association between μHC and λLC was dispensable for pentameric IgM formation during biosynthesis (Fig 14C, panels 1 and 5). The disruptive effect of C414S on pentameric IgM formation was negligible or null (compare panels 1 and 2; panels 5 and 6). This strongly demonstrates the differential effect of C414S mutation depending on the context of IgM formats. By contrast, the C575S and C414/575S mutants showed disruptive effects in increasing the ratio of monomeric IgM species but still retained the ability to maintain pentameric IgM species by non-covalent associations (panels 3 and 4; panels 7 and 8). The effect of C414/575S double mutant also showed markedly different disruptive effects between hexameric IgM and pentameric IgM.

These SEC results confirmed that depending on the location of the disulfide bond ablation within hexameric IgMs or pentameric IgMs, non-covalent inter-chain interactions can maintain the overall polymeric IgM product integrity (such as Cys-137 and Cys-337 in hexamers). Importantly, it became clear that mutations such as C414S, C575S, and C414/575S can produce a different spectrum of disruptive effects depending on whether it is introduced to hexameric IgM or pentameric IgM contexts. For example, the C414/575S double mutant was too disruptive for the non-covalent forces to sustain the hexamer IgM formation, but it was more tolerated in pentamer IgM assembly. These results illustrated that forces from non-covalent protein-protein associations assume different degrees of importance in holding the polymeric

IgM products together depending on whether the products are hexamers or pentamers during IgM biosynthesis and secretion.

## 4. Discussion

IgMs are megadalton-size secretory proteins that continue to attract cell biologists to investigate how they negotiate their abundant secretion with the ER quality control mechanism despite their structural complexity and biosynthetic burden [18, 40, 47, 49, 67]. To further probe the details of IgM assembly and secretion requirements, we holistically characterized the biosynthetic process of human IgM in a fully recombinant setting using a natural human IgM called SAM-6 as our model cargo, both in hexamer and pentamer contexts. By creating a series of mutant subunit constructs that block secretion and/or ablate the formation of specific inter-chain disulfide bonds, we tested the effects of mutations on IgM biosynthesis in 57 subunit mix-and-match combinations. Specifically, we investigated the effect of mutations on (1) protein synthesis and secretion output, (2) subunits' steady state subcellular distribution during biosynthesis, (3) induction of ER-associated inclusion bodies, (4) intracellular subunit's detergent solubility, (4) characterization secreted product quality by SDS-PAGE under non-reducing condition, and (5) secretory product quality assessment by analytical SEC under a physiological condition. Because we tried to understand the biosynthesis of one model human IgM systematically by combining the analysis of intracellular phenotypes and secreted product quality, this study consolidated and solidified a series of overlapping yet fragmented concepts on IgM assembly that accrued over the past few decades using various sources of IgMs, cellular models, and experimental systems. In addition, from the position of consolidated understanding, we uncovered unexpected underlying features related to IgM assembly and secretion.

### 4.1. Not all the inter-chain interactions need to be stabilized by disulfide bonds to assemble and secrete polymeric IgMs

One of the unexpected findings was the evidence that not all the prescribed inter-chain disulfide bridges were required for the polymeric IgM formation and secretion. Evidently, non-covalent protein-protein associations appear strong enough in some inter-chain interfaces to withstand the loss of pertinent disulfide bonds. A complete set of inter-chain disulfide formations, therefore, does not appear to be the most critical attribute for the ER quality control surveillance mechanism to assess the product quality and secretion readiness of assembling IgMs. The evidence showed that inter-chain interactions could be stabilized through non-covalent protein-protein interactions at least in two locations: (1) between μHC (Cys-137) and λLC (Cys-213) both for hexamers and pentamers, (2) between the pair of juxtaposing μHC monomers via Cys-337 at least for hexamers. Because C414S and C575S mutants did not completely lose the ability for covalent assembly of polymeric IgMs, the SEC results could not distinguish how much polymeric IgM formation was shared by covalent and non-covalent protein-protein associations. However, the roles of Cys-414 and Cys-575 were not redundant, as C414S and C575S showed varying degrees of disruptive effect on covalent polymeric IgM formation. Furthermore, the C414S point mutation, in particular, produced different severity on IgM assembly depending on whether the mutation was introduced in hexamers or pentamers. Importantly, the C414/575S double mutation revealed that non-covalent interactions alone were not strong enough to compensate for the simultaneous loss of two inter-chain disulfide bonds that cooperatively hold two adjacent IgM protomers together in hexameric IgM context. In stark contrast, the C414/575S double mutation was still partially tolerated to assemble pentameric IgM partly through non-covalent protein-protein interactions. Nonetheless, it is fascinating that polymeric IgM products can be partly held by non-covalent associations, and such

products can fulfill the ER quality control criteria while remaining associated during secretory pathway trafficking and even after the products are secreted. This feature highlighted the crucial roles of underlying non-covalent forces that not only orchestrate the initial subunit interactions but also maintain the polymeric IgM product integrity during and after secretion. The intricate IgM assembly process is actually far more robust than it appears and even equipped with a stop-gap measure that tolerates the secretion even when a certain set of inter-chain disulfide bond formation is incomplete. Such biosynthetic robustness may underwrite the abundant product secretion not only from a fully differentiated professional secretory plasma cell but also from non-professional secretory cells like HEK293 and others [42, 51, 53]. However, the side effects of missing a specific inter-chain disulfide bond between μHC and λLC on binding affinity as well as the consequence of missing another inter-chain disulfide between μHC monomers mediated by Cys-337 on cytolytic activity or epithelial transport are not yet understood.

While the dispensability of inter-chain disulfides seemed striking initially, it was not unusual in the realm of immunoglobulin evolution in vertebrate species. If we look beyond the human IgMs, there are many examples of normal immunoglobulins in which interactions between HC and LC are non-covalently maintained. For instance, in mouse IgA [68] and human IgA2 m(1) allotype [69, 70], a disulfide-linked LC dimer (LC–LC) and a disulfide-linked HC dimer (HC–HC) assemble non-covalently to produce a functional IgA without inter-chain disulfides between HCs and LCs. Similarly, IgA purified from chicken bile lacked the inter-chain disulfide bridges between HC and LC, and the IgA was shown to dissociate into αHC homodimers and free LC monomers without breaking the disulfide bonds [71]. Canine, equine, and porcine milk also contain IgAs that dissociate into HCs and LCs without disulfide reduction [71]. In an extreme case, four different mouse monoclonal IgA mAbs were shown to dissociate into free HC monomers and free LC monomers in SDS-PAGE under non-reducing conditions; and upon SDS removal, these monomers of HC and LC assembled back into functional IgAs that retained the antigen binding function [72]. In the studies on the primitive jawless vertebrates such as hagfish [73] and sea lamprey [74], antibodies seem to exist in 6.6S and 14S forms and behaved like monomeric and polymeric IgMs, respectively, but they lacked inter-chain disulfide linkage between HCs and LCs. Furthermore, in bullfrogs (*Rana catesbeiana*), free LC monomers are non-covalently associated with the disulfide-linked HC homodimers in all classes of immunoglobulins [75]. A similar feature was reported for the IgY of cane toad (*Bufo marinus*) [76]. There is yet another trick to do away with the inter-chain disulfide between HC and LC. In the μHC primary sequences of green anole lizard (*Anolis carolinensis*) and Chinese crocodile lizard (*Shinisaurus crocodilurus*), the critical Cys residue supposed to be present in the CH1 domain is absent [77, 78]. Moreover, in Chinese crocodile lizard, the LC sequence additionally lacked the critical C-terminal Cys residue [78] responsible for the inter-chain disulfide bonding. In this lizard species, the inter-chain disulfide bond formation between HC and LC is reciprocally blocked on both subunits as if making sure to thwart the formation of an inter-chain disulfide bond. As a result, the HC and LC of lizard polymeric IgM are held together only by non-covalent forces [77, 78]. Therefore, what we uncovered in this study for natural human IgM SAM-6 is evolutionarily and mechanistically prevalent in some branches of the animal kingdom or different classes of immunoglobulins.

## 4.2. A comprehensive list of secretory products and intermediates produced from different combinations of mutant subunit chains

This study revealed the production and release of various by products and assembly intermediates during the parental and mutant IgM expression. Interestingly, depending on the

mutations, some products were stringently retained in the cell due to the strict ER quality control mechanisms, while other products were secreted to the culture media abundantly. To holistically view the collection of products and by-products characterized in this study, we collated the snapshot information on product types and exhibited them as an inclusive reference chart (see S13 Fig). The chart shows (1) steady-state subcellular cargo distribution during over-expression, (2) the induced inclusion body types, if any, (3) the types of main secretory products and major by-products, (4) a set of common assembly intermediates released to the culture media, and (5) secretion output levels. We compiled this diagram hoping to serve as a reference to determine what can happen to IgM assembly and secretion when specific mutations are introduced to HCs and LCs and when a certain subunit chain expression is dropped out. We hope this study increases the awareness of IgM and IgM-like proteins as promising modality options for biotechnology applications by lowering the hurdle of working with recombinant IgMs.

## 4.3. Requirements for the specialized chaperones upregulated in IgM-producing plasma cells

Despite the intrinsic molecular complexity and the demand for cellular resources to assemble IgM molecules, an early study published in 1986 reported that plasma cells can produce and secrete ~25,000 IgM molecules per cell per second [19]. To explain the cellular feat like this, it has been postulated that plasma cells acquire enhanced secretory capacity by upregulating specialized ER resident chaperones during a differentiation process into a mature professional secretory cell to assist the assembly of IgM molecules [41–43, 79]. Interestingly, Cattaneo and Neuberger [51] showed that glioma, phaeochromocytoma, and other non-lymphoid cell lines were able to secrete polymeric IgM upon transfection of µHC and LC to a comparable level as plasmacytoma cell hosts. Likewise, Niles et al. [53] reported that IgM can be assembled and secreted by a murine AtT20 pituitary cell line regardless of developed or rudimentary secretory apparatus and without requiring specialized factors only available in plasma cells. Cell types of non-lymphoid origin are also sufficiently equipped to produce both pentameric and hexameric IgMs if the cells were transfected with µHC and λLC constructs with or without the JC subunit.

Now, because our model HEK293(6E) cell was also able to assemble and secrete both pentameric and hexameric forms of IgM, our study agreed that B-cell and plasma cell-specific ER resident proteins and chaperones were not stringently required for the formation and secretion of polymeric IgMs per se. It is still an intriguing possibility that plasma cell-specific factors play more critical roles in maximizing the success rate of polymeric IgM assembly while minimizing the release of assembly intermediates such as monomeric IgMs (µ2λ2) and half-molecules (µλ).

## 4.4. Diversity of ER-associated inclusion body phenotypes during IgM biosynthesis

Induction Russell body-like inclusion body by µHC and its suppression by LC co-expression during SAM-6 IgM biosynthesis agreed well with the previous findings from various IgG mAbs [33–39, 60]. µHC expression invariably induced Russell body-like globular, detergent-insoluble, aggregates even when the Cys residues in the constant region were mutated singly or doubly. Unless all four Cys residues participating in inter-chain disulfide bonding were mutated simultaneously, Russell body-like aggregate formation was not blocked, and detergent solubility and secretion were not restored. In this respect, Russell body formation was an unequivocal visual cue that reports the lack of cargo secretion and intracellular subunit

detergent insolubility. Likewise, the µHC-ΔCH1 mutant induced Russell body-like aggregates extensively. And µHC-ΔCH1 secretion remained suppressed until both Cys-414 and Cys-575 were mutated simultaneously, upon which the Russell body phenotype was dissolved, and the cargo secretion was restored. Similarly, the co-expression of the [µHC + JC] pair and [µHC-ΔCH1 + JC] pair led to the co-aggregation of both subunits in the ER and induced Russell body-like aggregation. Upon mutating the Cys-575 into Ser, JC became excluded from the Russell body and returned to its normal ER distribution.

The ΔCH1 version of µHCs has been widely used as a convenient model to induce Russell body phenotype [80, 81]. As shown previously by Corcos et al. [82], co-expression of cognate λLC prevented the Russell body formation of µHC-ΔCH1. The expression of cognate LC indeed exhibited such protective effects against Russell body formation in SAM-6. Because the λLC and µHC-ΔCH1 subunits do not covalently assemble, non-covalent inter-chain interactions between them must have exerted the effects to prevent the µHC-ΔCH1 aggregation into Russell body. We also found that the VL region protein alone can provide similar protection against µHC-ΔCH1 aggregation, indicating that non-covalent interactions between VL and VH domains are sufficient to discourage Russell body-like aggregation of µHC-ΔCH1. The interesting twist was that the protective effect of LCs on µHC-ΔCH1 aggregation was easily canceled when the JC subunit was also brought into the same assembly milieu in the ER (Fig 8F, first row). Because µHC-ΔCH1 and JC can interact covalently via a disulfide bond at Cys-575 (see Fig 7B and 7C), once these two proteins covalently aggregated into Russell body, an attempt to prevent the Russell body formation by non-covalent forces by λLC was no longer sufficient to prevent Russell body formation. Likewise, given the JC's tendency to form intra- and inter-chain disulfide bonds, it is likely that JC is playing cross-linking functions as previously suggested by Mattioli et al. [81]. Our work additionally demonstrated that C575S mutation can easily cancel the Russell body formation in the 3-chain expression of [µHC-ΔCH1 (C575S) + λLC + JC] (see Fig 8F, second row). It is likely that the covalent aggregation between JC and the Cys-575 of µHC-ΔCH1 overrode the protective benefits of λLC. Interestingly, the deletion of CH1 was required in this [µHC-ΔCH1 + λLC + JC] 3-chain Russell body formation event because a specific blockade of HC–LC inter-chain disulfide bond alone by a minimally invasive C137S mutation failed to induce Russell body phenotype in the [µHC (C137S) + λLC + JC] co-expression setting (Fig 10E top row), suggesting that a gross structural abnormality such as ΔCH1 is required for this particular aggregation event.

Whenever a secretion-competent polymeric IgM (regardless of hexamer or pentamer) was synthesized, the assembling IgM cargo was detected in the punctate structures distributed broadly in the cytoplasm. Only after the blockade of covalent polymerization by C414/575S did the punctate localizations disappear and the cargo distributed to the ER and Golgi. Interestingly, a poorly secreted cargo (e.g., [µHC-ΔCH1 + LC] pair and [µHC-ΔCH1 + VL] pair) also produced such cytoplasmic puncta to a similar extent as the parental subunit pair. We propose that punctate localization indicates that µHC or µHC-ΔCH1 is achieving a multimeric state but is independent of the cargo's secretion competence. Then, what are these cytoplasmic punctate structures that appear when polymeric IgMs are produced? Although the puncta showed a superficial resemblance to the ER exit sites, the co-staining experiment (S1 Fig) did not reveal their identity. Because both µHC and λLC consistently partitioned into soluble and particulate fractions equally in detergent solubility assay (S5 Fig), the punctate staining may reflect a pool of phase-separated, or condensed, IgM species that partitions into particulate fractions.

In addition to the Russell body-like aggregates, protein droplet inclusion bodies were also induced during IgM expression when the JC subunit was co-expressed with µHC (C414/575S) or µHC-ΔCH1 (C414/575S) subunits (Fig 7D and S6C Fig). While µHC (C414/575S) mutant

induced Russell body when it was expressed alone, the co-expression of JC led to such a dramatic change in inclusion body morphology and property. Similarly, although µHC-ΔCH1 (C414/575S) localized to ER and Golgi at steady state, JC co-expression changed the solution behavior of µHC-ΔCH1 (C414/575S) mutant to induce protein droplet inclusion bodies. Instead of being present in the ER as an inert bystander, JCs interacted with these mutant µHC subunits non-covalently and modulated their condensation propensity. Unlike the previously characterized scFv-Fc-stp [60], we did not readily find evidence suggesting µHC (C414/575S) and µHC-ΔCH1 (C414/575S) have solubility problems at high concentrations or cryoglobulin-like characteristics at temperatures below 37˚C. The mechanistic process of how JC facilitated the droplet inclusion body formation was elusive. Whether or not LLPS played roles in this droplet formation was also unknown.

While these protein droplets can be readily distinguished from the classical detergent-insoluble Russell body-type inclusions by using DIC and immunofluorescence microscopy, there is some confusion in scientific and clinical literature in that both types of inclusion bodies are referred to as Russell bodies indiscriminately. Furthermore, there is yet another type of ER-associated inclusion body composed of immunoglobulin crystals [34, 35, 37, 83] or LC crystals [34, 39]. Although our model IgM SAM-6 did not have such crystallizing propensity, there are numerous reports on intra-ER protein crystals composed of IgMs [84–87]. Because intrinsic physicochemical properties of individual immunoglobulins appear to dictate the solubility, condensation, and the type of ER-associated inclusion bodies they induce [33–39, 60, 83], the formation of different types of ER-associated inclusion bodies may just be the variations of a common theme. A precise understanding of the relationships between individual immunoglobulin's physicochemical properties and their intra-ER condensation propensity is awaited for effective disease diagnosis and antibody biotherapeutics manufacturing. Lastly, in the age of diversity and inclusion, we suggest that different types of ER-associated immunoglobulin inclusion bodies can be called Russell bodies regardless of their subunit composition, detergent solubility, morphology, or PAS staining.

## Supporting information

**S1 Fig. IgM-positive punctate structures do not co-localize with representative organelle markers of the secretory and endocytic pathways.** Fluorescent micrographs of HEK293 cells transfected with the [µHC + λLC] construct pair. On day-3 post-transfection, cells were fixed, permeabilized, and co-stained with a 1-to-1 mix of FITC-labeled anti-µHC and FITC-labeled anti-λLC to stain both subunits simultaneously (green) and antibodies against various organelle markers (red). Green and red image fields were superimposed to create 'merge' views. (TIF)

**S2 Fig. µHC-induced Russell body-like globular aggregates co-localize with ER-resident and ER-associated proteins.** Fluorescent micrographs of HEK293 cells transfected with the µHC construct. On day-3 post-transfection, cells were fixed, permeabilized, and co-stained FITC-labeled anti-µHC (green) and antibodies against various organelle markers (red). Green and red image fields were superimposed to create 'merge' views. (TIF)

**S3 Fig. Russell body-like globular aggregates induced by µHC-ΔCH1 mutant co-localize with ER-resident and ER-associated proteins.** Fluorescent micrographs of HEK293 cells transfected with the µHC-ΔCH1 mutant construct. On day-3 post-transfection, cells were fixed, permeabilized, and co-stained FITC-labeled anti-µHC (green) and antibodies against various organelle markers (red). Green and red image fields were superimposed to create

'merge' views.
(TIF)

**S4 Fig. The isolated VL region alone can prevent the Russell body-like aggregation of µHC-ΔCH1 construct.** (A) Schematic representation of the VL-only construct. (B, C) HEK293 cells were transfected with VL-only construct (lanes 1, 3, 5) or the full-length λLC (lanes 2, 4, 6). On day-7 post-transfection, cell lysates (lanes 1 and 2) and cell culture media samples (lanes 3–5) were prepared and resolved by SDS-PAGE under reducing or non-reducing conditions followed by Coomassie blue staining (B) or by Western blotting (C). The membrane was probed with polyclonal anti-λLC. The corresponding protein band for the VL-only protein is pointed and labeled in panel B, lanes 3 and 5. Monomeric and dimeric λLC subunit is also labeled next to lane 6. The polyclonal anti-λLC raised against the constant domain of λLC could not recognize the VL-only protein in Western blotting. (D, E) Fluorescent micrographs of HEK293 cells transfected with the constructs shown on the left side. On day-3 post-transfection, cells were fixed, permeabilized, and co-stained with FITC-labeled anti-µHC and Texas Red-labeled anti-λLC. The polyclonal anti-λLC was raised against the constant domain of λLC and could not recognize the VL-only protein.
(TIF)

**S5 Fig. Detergent solubility of intracellular subunit chains under different co-expression settings.** (A) On day-3 post-transfection, the detergent solubility was determined for the intracellular pool of transfected µHC and/or λLC subunit. The total detergent cell extracts were prepared under non-denaturing conditions and subjected to 15,000 g centrifugation for 60 min. Then, total (T), soluble (S), and particulate (P) fractions were resolved in SDS-PAGE under reducing conditions. Transfected construct(s) are shown at the top of corresponding lanes. Membranes were probed with polyclonal anti-IgM (H+L) (top panel) and monoclonal anti-GAPDH (second panel). Unidentified, non-specifically cross-reacting proteins are also shown in the bottom panel as the reference for soluble and particulate proteins. (B) The effect of co-expression partners on the detergent solubility of JC subunit was determined. Membranes were probed with polyclonal anti-IgM (H+L) (top panel), monoclonal anti-JC (middle panel), and monoclonal anti-GAPDH (third panel).
(TIF)

**S6 Fig. Intracellular J-chain distribution is influenced by the Cys-575 of the µHC-ΔCH1 subunit.** Fluorescent micrographs of HEK293 cells co-transfected with JC and one of the following constructs: (A) µHC-ΔCH1, (B) µHC-ΔCH1 (C575S), or (C) µHC-ΔCH1 (C414/575S). Co-transfected construct pairs are also shown on the left side of each row. On day-3 post-transfection, cells were fixed, permeabilized, and co-stained with FITC-labeled anti-µHC and monoclonal anti-JC (shown in red). (D) On day-7 post-transfection, cell culture media (lanes 1–3; lanes 7–9) and cell lysates (lanes 4–6) were analyzed by Western blotting. Membranes were probed with polyclonal anti-IgM (H+L). Co-transfected construct pairs are shown at the top of each lane. (E) The same culture media and cell lysate samples were analyzed by Western blotting using monoclonal anti-JC. A longer exposed Western blot result is shown underneath the corresponding lanes in a black box for the cell culture media (panel E, lanes 1–3).
(TIF)

**S7 Fig. The penultimate Cys-213 residue of λLC is required for inter-chain disulfide bond formation.** (A, top) Schematic representation of the full-length SAM-6 λLC (top row) and its ΔCS mutant (second row) in which two C-terminal amino acids (Cys-213 and Ser-214) are deleted. (A, bottom) The position of Cys-213 residue involved in the HC–LC inter-chain disulfide bond is highlighted in yellow in the context of hexameric IgM. Solid red lines represent

the inter-chain disulfide bond connectivity. (B, C) HEK293 cells were transfected with full-length λLC (lanes 1, 3, 5) or its ΔCS mutant (lanes 2, 4, 6). On day-7 post-transfection, cell lysates (lanes 1 and 2) and cell culture media samples (lanes 3 and 4) were prepared and resolved by SDS-PAGE under reducing conditions followed by Coomassie blue staining (B) or by Western blotting (C). The day-7 cell culture media were also analyzed by Coomassie staining or Western blotting after resolving the proteins under non-reducing conditions (B C, lanes 5 and 6). Membranes were probed with polyclonal anti-λLC. The corresponding protein band for the λLC subunit is pointed by an arrowhead and labeled. Monomeric and dimeric λLC subunit is labeled next to lane 6. (D, E) Fluorescent micrographs of HEK293 cells transfected with full-length λLC (D) or ΔCS mutant (E). On day-3 post-transfection, cells were fixed, permeabilized, and co-stained with FITC-labeled anti-CD147 and Texas Red-labeled anti-λLC. (TIF)

**S8 Fig. Effect of C137S and C337S point mutations on the expression, secretion, and subcellular localization of μHC subunit.** (A, left) Schematic representation of parental SAM-6 μHC (top row) and its C137S and C337S mutants (second and third rows). (A, right) The position of Cys-137 and Cys-337 residues is highlighted in yellow in the context of hexameric IgM. Solid red lines represent the inter-chain disulfide bond connectivity. (B, C) HEK293 cells were transfected with parental μHC and its mutants, as shown at the top of each lane. On day-7 post-transfection, cell culture media (B) and cell lysates (C) were prepared and resolved by SDS-PAGE under reducing conditions followed by Coomassie blue staining (B, C, left panels) or by Western blotting (B, C, right panels). Membranes in B and C were probed with polyclonal anti-IgM (H+L). Both parental and mutant μHCs were completely retained in the cells and failed to secrete. (D) Fluorescent micrographs of HEK293 cells transfected with parental μHC (top row), μHC (C137S) mutant (second and third rows), or μHC (C337S) mutant (fourth and fifth rows). On day-3 post-transfection, cells were fixed, permeabilized, and co-stained with FITC-labeled anti-CD147 and Texas Red-labeled anti-μHC. (TIF)

**S9 Fig. Co-aggregation of J-chain with μHC (C137S) and μHC (C337S) mutants.** (A–C) HEK293 cells were co-transfected with JC and one of the following μHC constructs: parental μHC (lanes 1, 4, 7), μHC (C137S) (lanes 2, 5, 8), and μHC (C337S) (lanes 3, 6, 9). On day-7 post-transfection, cell lysate samples were prepared (lanes 1–3), and cell culture media were harvested (lanes 4–9) to run SDS-PAGE under reducing conditions (lanes 1–6) or non-reducing conditions (lanes 7–9) followed by Coomassie blue staining (panel A) and Western blotting (panels B and C). Membranes were probed with (B) polyclonal anti-IgM (H+L) or (C) monoclonal anti-JC. A co-transfected construct pair is shown at the top of each lane. (D) Fluorescent micrographs of HEK293 cells co-transfected with JC and parental μHC (top row), JC and μHC (C137S) (second row), and JC and μHC (C337S) (third and fourth rows). On day-3 post-transfection, cells were fixed, permeabilized, and co-stained with FITC-labeled anti-μHC (green) and monoclonal anti-JC (red). (TIF)

**S10 Fig. Effects of μHC (C137S), μHC (C337S), μHC (C414S), μHC (C575S), μHC (C414/ 575S), and μHC (4×C>S) mutation on overall IgM protein expression and secretion.** (A–C) The effect of μHC (C137S) mutant subunit on IgM expression and secretion was assessed both in a 2-chain co-expression (lanes 1–4) and a 3-chain co-expression (lanes 5–8) settings. The DNA ratio of each subunit chain is shown at the top of each lane. On day-7 post-transfection, cell culture media (A, B) and cell lysates (C) were resolved by SDS-PAGE under reducing conditions followed by Coomassie blue staining (A) or Western blotting (B, C). Membranes

were probed with polyclonal anti-IgM (H+L) to detect μHC or μHC (C137S) (top panel) and λLC or λLC-ΔCS (second panel) or with monoclonal anti-JC (third panel). (D–F) The effect of C337S mutation on IgM expression was tested both in a 2-chain (lanes 1–3) and a 3-chain co-expression (lanes 4–6) settings. The DNA ratio of each subunit chain is shown at the top of each lane. On day-7 post-transfection, cell culture media (D, E) and cell lysates (F) were resolved by SDS-PAGE under reducing conditions, followed by Coomassie blue staining and Western blotting. Membranes were probed with polyclonal anti-IgM (H+L) to detect μHC or μHC (C337S) (E, F; top panel) and λLC or λLC-ΔCS (E, F, second panel) or with monoclonal anti-JC (E, F, third panel). (G–J) The effect of μHC (C414S) mutant on polymeric IgM expression was tested both in a 2-chain (panel G, H) and 3-chain co-expression (panel I) settings. To detect any subtle effects, μHC (C414S) was compared side by side with μHC (C575S) and μHC (C414/575S). On day-7 post-transfection, cell culture media (H, J) and cell lysates (G, I) were resolved by SDS-PAGE under reducing conditions, followed by Coomassie blue staining (H) and Western blotting (G, I, J). Membranes were probed with polyclonal anti-IgM (H+L) to detect μHC variants and λLC variants or monoclonal anti-JC (I, J; bottom). (K–M) The effect of μHC (4×C>S) mutant on IgM expression and secretion was tested both in a 2-chain (lanes 1–4) and a 3-chain co-expression (lanes 5–8) settings. The DNA ratio of each subunit chain is shown at the top of each lane. On day-7 post-transfection, cell culture media (K, L) and cell lysates (M) were resolved by SDS-PAGE under reducing conditions, followed by Coomassie blue staining (K) and Western blotting (K, M). Membranes were probed with polyclonal anti-IgM (H+L) to detect μHC or μHC (4×C>S) (L, M; top panel) and λLC or λLC-ΔCS (L, M; second panel) or with monoclonal anti-JC (L, M; third panel).
(TIF)

**S11 Fig. Effect of the C414S point mutation on the expression, secretion, and subcellular localization of μHC subunit.** (A–C) HEK293 cells were transfected with the construct or construct pair shown at the top of each lane. On day-7 post-transfection, cell culture media (lanes 5–12) and cell lysates (lanes 1–4) were resolved by SDS-PAGE under reducing (lanes 1–8) or non-reducing (lanes 9–12) conditions followed by Coomassie blue staining (A) or by Western blotting (B, C). Membranes in B and C were probed with polyclonal anti-IgM (H+L) and monoclonal anti-JC, respectively. (D) Fluorescent micrographs of HEK293 cells transfected with parental μHC (top row) or μHC (C414S) mutant (second row). On day-3 post-transfection, cells were fixed, permeabilized, and co-stained with FITC-labeled anti-CD147 and Texas Red-labeled anti-μHC. (E) Fluorescent micrographs of HEK293 cells transfected with [μHC + JC] pair (top row) or [μHC (C414S) + JC] pair (second row). Cells were co-stained with FITC-labeled anti-μHC (green) and monoclonal anti-JC (shown in red).
(TIF)

**S12 Fig. Effect of μHC (4×C>S) quadruple mutations on the expression, secretion, and subcellular distribution of μHC subunit.** (A) Schematic representation of parental SAM-6 μHC (top row) and its 4×C>S mutant (second row). (B) The position of all Cys residues involved in inter-chain disulfide bond formation on μHC and λLC are depicted in the context of hexameric IgM. Solid red lines represent the inter-chain disulfide bond connectivity. (C, D) HEK293 cells were transfected with parental μHC alone (lane 1) or its 4×C>S mutant alone (lane 2). Likewise, the cells are co-transfected with [μHC + JC] pair (lane 3) or [μHC (4×C>S) + JC] pair (lane 4). On day-7 post-transfection, cell lysates (C) and culture media (D) were resolved by SDS-PAGE under reducing conditions followed by Coomassie blue staining (C, D; left panel) or by Western blotting (C, D; right panels). Membranes in C and D were probed with polyclonal anti-IgM (H+L) (top panel) or monoclonal anti-JC (bottom panel). (E) Day-7 culture media were also analyzed by Western blotting after proteins were resolved under non-

reducing conditions. Membranes were probed with polyclonal anti-IgM (H+L) (left panel) or monoclonal anti-JC (right panel). (F) Fluorescent micrographs of HEK293 cells transfected with parental μHC (top row) or μHC (4×C>S) mutant (second and third rows). On day-3 post-transfection, cells were fixed, permeabilized, and co-stained with FITC-labeled anti-CD147 and Texas Red-labeled anti-μHC. (G) Fluorescent micrographs of HEK293 cells co-transfected with [μHC + JC] pair (top row) or [μHC (4×C>S) + JC] pair (second and third rows). Cells were co-stained FITC-labeled anti-μHC (green) and monoclonal anti-JC (shown in red).
(TIF)

**S13 Fig. Diagrams of covalently assembled secretory products, by-products, and assembly intermediates under different subunit combinations.** The types of secreted main products and significant by-products released to culture media are illustrated. (A) single-construct expression setting. (B) 2-chain co-expression setting. (C) 3-chain co-expression setting. The name of the transfected construct (or a set of constructs) is shown in blue letters at the upper-most area of each box. At the lower-most area, common types of assembly intermediate in each condition are shown using the numbering system categorized in panel D. Steady state subcellular distribution of individual subunits for each transfection setting is shown in the space under each box, in green letters. When there is no mention of the secretion outputs, it suggests that the products were secreted abundantly. (D) Six commonly produced assembly intermediates released to the culture media are illustrated in each box, from 1 to 6. Solid red lines represent the inter-chain disulfide bond connectivity.
(TIF)

**S1 Raw images.**
(PDF)

## Acknowledgments

The authors thank Christy Tinberg for analyzing the SAM-6 VL and VH primary sequences for potential abnormality. We thank our colleagues at SARC for an initial attempt to optimize the hexamer IgM purification method. HH is personally grateful to Yoko Azumi for her continuous encouragement.

## Author Contributions

**Conceptualization:** Haruki Hasegawa.

**Data curation:** Haruki Hasegawa, Songyu Wang, Eddie Kast, Hui-Ting Chou.

**Formal analysis:** Haruki Hasegawa.

**Investigation:** Haruki Hasegawa, Songyu Wang, Eddie Kast, Hui-Ting Chou.

**Methodology:** Haruki Hasegawa.

**Project administration:** Haruki Hasegawa.

**Resources:** Haruki Hasegawa, Songyu Wang, Mehma Kaur, Tanakorn Janlaor, Mina Mostafavi, Yi-Ling Wang, Peng Li.

**Supervision:** Haruki Hasegawa.

**Validation:** Haruki Hasegawa.

**Visualization:** Haruki Hasegawa.

Writing – **original draft:** Haruki Hasegawa.

Writing – **review & editing:** Haruki Hasegawa, Songyu Wang, Peng Li.

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
