## [Decision Letter · Decision Letter 0]

2 Feb 2024

PONE-D-23-28036Understanding the biosynthesis of human IgMs through a combinatorial expression of mutant subunits that affect different assembly steps.PLOS ONE

Dear Dr. Hasegawa,

Thank you for submitting your manuscript to PLOS ONE. After careful consideration, we feel that it has merit but does not fully meet PLOS ONE’s publication criteria as it currently stands. Therefore, we invite you to submit a revised version of the manuscript that addresses the points raised during the review process.

 1) While both reviewers appreciate the enormous amount of data generated in this study, it overwhelms any reader.  I strongly recommend to focus on the novel findings, and to condense the confirmatory data into supplemental data/figures.    The whole manuscript should be streamlined and significantly shortened.   2) Both reviewers are also concerned about the discussion of some seemingly controversial data that could simply be explained by differences in the experimental systems.  These extensive discussion also distract from the main novel findings and I suggested to condense respective sections. 3) Address all the concerns of the reviewers by making appropriate changes to the text and provide additional experimental data if required to support the current conclusions.  .     

We look forward to receiving your revised manuscript.

Kind regards,

Sebastian D. Fugmann, Ph.D.

Academic Editor

PLOS ONE

Journal Requirements:

4. We notice that your supplementary figures are included in the manuscript file. Please remove them and upload them with the file type 'Supporting Information'. Please ensure that each Supporting Information file has a legend listed in the manuscript after the references list.

Reviewers' comments:

Reviewer's Responses to Questions

**Comments to the Author**

1. Is the manuscript technically sound, and do the data support the conclusions?

Reviewer #1: Yes

Reviewer #2: Partly

2. Has the statistical analysis been performed appropriately and rigorously? 

Reviewer #1: Yes

Reviewer #2: N/A

3. Have the authors made all data underlying the findings in their manuscript fully available?

Reviewer #1: Yes

Reviewer #2: Yes

4. Is the manuscript presented in an intelligible fashion and written in standard English?

Reviewer #1: Yes

Reviewer #2: Yes

5. Review Comments to the Author

Reviewer #1: The authors present a very thorough study on IgM assembly. A large number of mutants have been studied and the findings obtained largely support current models for IgM assembly. The new findings are buried in the impressive data set. While it is of value to repeat previous findings as controls, the new insight needs to be better highlighted. All in all the manuscript is a very long read and therefore should be shortened. The story and main message is missing/unclear. Many of the data is a reproduction of previous findings. Some aspects are new and interesting – the authors should elaborate more on them. Some conclusions do not seem to be correct and should be revisited (chapter 3.12).

In general, the manuscript would benefit from shortening and reorganizing.

Specific points

1. Mutants that confirm previous findings should be summarized in the supplementary information.

2. The text needs to be shortened. Often experiments and results are described in too much detail and length.. For example, one could state that μHC expressed alone always formed Russel bodies regardless of mutations – with the exception of 4x C>S. Adescription of each mutant in this context is overwhelming if no additional effects are visible.

3. 3.1 - The first paragraphs are meticulously performed proofs of principle. I suggest shifting these to the supplements, as they do not contain new insights into the topic of IgM biosynthesis and assembly. Figure 1 and 2 belong in the supplement section.

The same is true for 3.2. Please highlight the one difference you spotted when co-expressing μ-HC-ΔCH1 with λLC. This information gets lost in the presence of the list of already known facts. The corresponding figures should be moved to the supplement. Please sho only show images for new findings.

4. The finding that Russel bodies are not formed by μ-HC-ΔCH1 C414/575S, which also is secretion competent, should be highlighted.

5. Please refer to this paper: https://doi.org/10.15252/embj.2021108518 in the context of chapter 3.5

6. The titration of JC DNA and the correlation with the amount of pentamer is a very nice experiment and should get way more spotlight in the context of this paper.

7. It is interesting, that you were able to secrete JC in the absence of μHC and λLC – something that was not reported before. This finding is worth more investigation.

8. 3.7 - The presence of pentamers and hexamers has been already published.

9. 3.8 - This section needs to be shortened and many data should be presented as supplementary information

10. The new finding about JC as part of Russel bodies depending on the presence of other chains is interesting. When CH1 is deleted, the effects stay the same. The evidence could be moved to the supplement and the text shortened.

11. 3.9 - It is already known, that C575 and C414 are necessary for correct pentamer assembly. C575 has the more important role. For example Hiramoto et. al. (DOI: 10.1126/sciadv.aau1199) already showed that for the C414S mutation, in addition to ‘loose’ pentamers, also tetramers can be formed. The cause, why the oligomers are not formed in your setting is probably the absence of C575.

12. 3.10 - The results with ΔCS mutant are very interesting. However, the length of the description is too long.

13 3.11 - This section could be combined with the previous chapter, where you state that the covalent link between CH1 and LC is not strictly necessary for hexamer/pentamer formation. This section is also too long. The summaries at the end of each chapter seem sufficient to present the data.

14. 3.12 - The C337S mutant was described previously in several publications; e.g. Wiersma et. al. described formation of hexamers and pentamers with this mutant, which of course dissociate on SDS-PAGE (https://doi.org/10.4049/jimmunol.160.12.5979). This is also reasonable, as single Cμ4tp domains are able to form hexamers in vitro (https://doi.org/10.1073/pnas.1701797114), which of course can only be observed with non-denaturing approaches like SEC. Most likely, the μ2λ2JC and μ2JC are part of such a non-covalent pentamer, which is dissociated by SDS-PAGE. The explanation proposed in the manuscript seems unlikely. Analysis of this mutant in the presence of LC AND JC by SEC seems needed. Formation of hexamers without JC is seen in Figure 14, chromatogram 6 but there are no intermediates between the monomer and hexamer.

15. 3.13 - The last paragraph of this chapter describes the effect of the 4x C>S mutant well (line 1177-1180), but again the text needs to be shortened.

16. 3.14 - Please show all the chromatograms together with the SDS gels of the more important mutants.

Reviewer #2: In an extensive tour de force for both writers and readers (>100 pages, almost 30,000 words, 15 multipaneled main figures and 7 supplemental ones) Hasegawa et al. investigate the biogenesis of multimeric IgM. They address this interesting problem analyzing systematically a panel of heavy and light chain mutants, that yield 48 possible combinations.

While the biochemical analyses are clear and complete, the imaging studies (mainly immunofluorescence) leave room for improvement and clarification.

However, besides some technological issues that can be addressed with additional work, the main criticism that can be raised to this extensive study is its largely confirmatory nature.

That interchain H-L bonds are not needed for antibody assembly is known since over 50 years. In many species IgA exist that lack them. And monoclonal antibodies endured clonal selection without a covalent association between their H and L chains.

Large parts of the wordy text is devoted to refuting papers published by others rather than to highlighting novel findings. Nothing wrong with the desire to clarify potential controversies. It seems to this reviewer, however, that the authors see controversies where there are simply different experimental systems. In doing so, the authors do not address the potentially relevant reasons that underlie different phenotypes in different cells expressing different proteins.

That said, the wealth of data and thoroughness of this systematic approach would make a profoundly revised manuscript suitable for publication.

Below follows a list of specific points that the authors are expected to carefully address before this manuscript can be reconsidered for publication.

Major points

The text should be shortened drastically. The main points of the study can be summarized in much less than one third of the actual size of the text and iconography. Many of the main figures can become supplemental information. For example, the authors ‘validate’ the efficiency of transfection in most experimental set ups. Validation is obviously essential, but the data can be easily summarized once in supplementary material. As presented, the paper reads like a lab book: the authors should make the effort to pick the relevant information and focus the reader’s attention on the novel aspects of their diligent work.

In general, condensation in RBs correlates with the presence of abundant detergent insoluble material. The authors may wish to investigate further this problem and discuss why no such ‘precipitates’ be formed in their system.

Higher resolution imaging are needed for some key mutants analyzed (below some detailed suggestions)

-Page 2, line 56. In view of the existing literature, it is not at all unexpected that the H-L and inter-µ chain bonds linking C337 be disposable for polymerization. That C414 and C575 can mediate different bonds was also already shown (Giannone et al., 2022 and references therein). What is solid in this study is the ‘holistic’ analysis in a single human IgM. The authors should rewrite the Ms focusing on the few novel findings and highlighting the systematic confirmation of what was already known from different models.

-Page 15 , line 344 and Figure 1 c-d. The distribution of free µHC observed by immunofluorescence (IF) cells differs from other reports in which unassembled H chains were shown to distribute in the ER, colocalizing with BiP or other markers of the organelle. Therefore, identification of the red blobs containing µHC as Russell bodies is premature. Calnexin costaining (shown in Figure 4) is not consistent with a RB, as defined by the accumulation of amorphous Ig aggregates in an intracellular compartment. A membrane protein like calnexin would be expected to be excluded from the lumen of ER-localized RB.

Are these µHC detergent insoluble, as most condensed-aggregated proteins in Mott cells are?

The authors should use other organelle markers and perform electron microscopy and exclude the possibility that aggresomes be formed by their transfectants.

In lymphoid cells unassembled µ and J chains are dislocated to the cytosol and degraded very rapidly by proteasomes (Chillaron et al, 2000: Mancini et al, 2000; Fagioli et al 2001), but more slowly in HeLa (Medrano-Fernandez et al, 2014). The authors should compare the rate and mode of the degradation as well as the solubility of µHC and µHC-∆CH1 before concluding that both form similar structures when overexpressed in the absence of L chains.

In general, it would be nice to see bigger enlargements of the IF images, and more markers (ERGIC and Golgi subregions) to better appreciate the localization of the key mutants (not all of them, the authors should try to focus the reader’s attention on the ones that give unexpected results).

-p18, line 395. The authors should discuss why SAM µHC behave differently from other H chains (VH? Human vs mouse? Etc).

-p20 line 453. The authors describe these ‘cytoplasmic puncta’, without characterizing them. In the discussion, they raise the possibility of them being ER exit sites. These structures should be characterized, using additional markers, ideally by CLEM.

-pages 20-21; the authors go a long way to refute some ‘influential papers’ published on the biogenesis of RBs. In Ref. 69, Mattioli et al. use HeLa transfectants to show a role of ERGIC in catalyzing the condensation of murine µHC∆CH1 chains, possibly by increasing the local concentration above a critical point. They also show a role of L chains in determining the site of condensation (smooth RBs in ERGIC, no L chains and rough RBs, ribosome coated ER, with L chains). A reasonable interpretation was that the tendency of CL domains to form homodimers facilitate condensation. Accordingly, a VL domain alone did not induce rRB formation. Would expression of a VL only suffice to change the phenotypes (see also p25)?

The authors should consider that not only cell-specific factors (ERGIC53, ERp44, the redox pose of the ER etc., see Ronzoni et al. 2010), but also subtle determinants in other domains, mainly the Variable ones, would dictate the rate, extent and localization of condensation/precipitation. These points should be further analyzed, as the differences between experimental systems may reveal novel phenomena, or at least constructively discussed.

P21, line 462. The potential ole of JC in favoring µHC-∆CH1 is of interest. Considering its tendency to form intra- and inter-chain bonds (see the smear secreted by HEK transfectants) JC may serve a ‘cross-linking’ role as described for L chain by Mattioli et al (2006).

-p25, line 555. The authors should consider expressing a VL alone or swapping VH domains. What is their role in catalyzing aggregation?

-p27, line597. It is of interest that JC alone did not form RBs but was trapped in ‘RB-like’ structures when µHC were there. Are JC degraded more rapidly when not trapped in RBs?

-p35 line 761. The nature and significance of these droplets should be explained and discussed.

-p36 line 791. Why is the JC smear no longer visible?

-p37 line 825. There are many things that one does not realize when submitting a manuscript. Papers are published if they contain original and important observation(s), like for instance that the absence of the BiP-binding CH1 is generally sufficient to induce intracellular condensation of immunoglobulins. The original observations are then confirmed (e.g. the Bruggeman lab showing that mice lacking L chains have Mott cells plenty of Ig lacking a CH1) and extended by the observation that J chains are trapped in µHC and µHC-∆CH1 ‘RBs’. Are JC detergent insoluble? Are JC trapped in IgG-containing RBs?

A potentially interesting point to follow up would be to determine whether µHC form mixed polymers with µHC-∆CH1 and are trapped in RBs. It remains possible that the two end up in different blobs.

-p38 line 827. The authors state ….it was clear that covalent aggregation between JC and µHC-∆CH1 .. was the underlying… the logic of this conclusion is not clear. As the authors elegantly confirm, JC are not needed to form RBs.

-p40 line 875. Sec MALS or sucrose gradient analyses should be described here to sustain the case. By the way, that the inter H-L bond is not necessary was known for 5 decades.

-p42 line 912. Can the authors exclude that SS reshuffling occurs after secretion.

An interesting endeavor -also considering the long discussion about the potential medical use of IgM variants- would be to compare the stability of the mutants with or without interchain bonds.

-p52 line 1133. The data suggest that unassembled µHC avoid BiP quality control ad are secreted. How do the authors explain this unexpected finding?

Some sections of the discussion sounds more like parts of a grant. Although probably tenable, some speculations are not sustained by data or elements that derive from this paper and could be deleted.

Minor points

-p16 line 365. Hexamers represent the vast majority of secreted IgM. Wouldn’t ELISA allow to quantify them?

-p17, lines 375 and following. Many previous studies showed that IgM can be assembled and secreted by non-lymphoid cells. Of interest, HeLa variants were isolated that failed to do so and secreted also non polymeric IgM subunits. Retention of intermediates was partially rescued by ERp44 and ERGIC53 overexpression, suggesting a defect in distal quality control. This is another example of how different cell sublines handle proteins differently. The unexpected can lead to novel findings, without the need to think that others made mistakes.

-p19 line 423: the sentence ‘…induces a spontaneous…’ sounds somehow contradictory

-p23, line 500 and following. The dual role of CH1-BiP and C575-ERp44 was shown over 30 years ago and confirmed in many further studies.

6. PLOS authors have the option to publish the peer review history of their article (what does this mean?). If published, this will include your full peer review and any attached files.

Reviewer #1: No

Reviewer #2: No

---

## [Author Response · Author response to Decision Letter 0]

31 Mar 2024

"Response to Reviewer" document is attached as a separate document. I will also copy and paste the same contents below, although the formatting will be lost. Thank you.

PONE-D-23-28036

Understanding the biosynthesis of human IgMs through a combinatorial expression of mutant subunits that affect different assembly steps.

PLOS ONE

Dear Dr. Hasegawa,

Thank you for submitting your manuscript to PLOS ONE. After careful consideration, we feel that it has merit but does not fully meet PLOS ONE’s publication criteria as it currently stands. Therefore, we invite you to submit a revised version of the manuscript that addresses the points raised during the review process.

1) While both reviewers appreciate the enormous amount of data generated in this study, it overwhelms any reader. I strongly recommend to focus on the novel findings, and to condense the confirmatory data into supplemental data/figures. The whole manuscript should be streamlined and significantly shortened. 

We shortened the text from ~30,000 words to ~23,000 words (including the Supplemental legends and references), although we added several new data figures requested by both reviewers. We also used Supplemental materials more effectively. Because IgMs are often considered a very complex group of secretory glycoproteins, we wanted to ensure we established a suitable experimental platform so that we could start asking questions on IgM biosynthesis to uncover something novel. It was a natural process to build upon many of the known important concepts that laid the foundation of IgM assembly mechanisms. This is an important thing to do, as Reviewer #2 pointed out that “differences in employed expression systems can play roles in revealing unexpected novel phenomena.” Instead of treating the data as “confirmatory” and sending them to Supplement, I decided to keep some of them in the main Figures.

It was also important to show all the cards because the cell line we used is highly relevant in antibody drug discovery research and for antibody therapeutics production in the biotech industry in general.

2) Both reviewers are also concerned about the discussion of some seemingly controversial data that could simply be explained by differences in the experimental systems. These extensive discussion also distract from the main novel findings and I suggested to condense respective sections.

As suggested, we removed unnecessary arguments and discussions when explaining some conflicting results. 

3) Address all the concerns of the reviewers by making appropriate changes to the text and provide additional experimental data if required to support the current conclusions. 

We performed detergent solubility analysis for the Russell body-like aggregates as suggested by Reviewer #2 (see Suppl. 10). We also performed two additional IF imaging studies. (1) Characterization of the IgM-positive punctate structures by co-staining with ERGIC and ER exit site markers (see Suppl. 1). (2) Characterization of Russell body-like aggregates by co-staining with additional ER resident makers such as BiP, ERp57, and calreticulin, as well as cis- and trans-Golgi markers such as giantin and p230 (see Suppl. 2 and Suppl. 3).

Regrettably, due to our organizational technical limitations, we were not able to perform one suggested transmission EM and CLEM study to characterize the IgM-containing cytosolic puncta.

We look forward to receiving your revised manuscript.

Kind regards,

Sebastian D. Fugmann, Ph.D.

Academic Editor

PLOS ONE

Journal Requirements:

The “data not shown” issue was addressed by actually showing the data as the new figure sets, as shown in Fig. 12 and Suppl. 11.

4. We notice that your supplementary figures are included in the manuscript file. Please remove them and upload them with the file type 'Supporting Information'. Please ensure that each Supporting Information file has a legend listed in the manuscript after the references list.

We followed these instructions.

Reviewer #1: 

The authors present a very thorough study on IgM assembly. A large number of mutants have been studied and the findings obtained largely support current models for IgM assembly. The new findings are buried in the impressive data set. While it is of value to repeat previous findings as controls, the new insight needs to be better highlighted. All in all the manuscript is a very long read and therefore should be shortened. The story and main message is missing/unclear. Many of the data is a reproduction of previous findings. Some aspects are new and interesting – the authors should elaborate more on them. Some conclusions do not seem to be correct and should be revisited (chapter 3.12). In general, the manuscript would benefit from shortening and reorganizing.

Specific points

1. Mutants that confirm previous findings should be summarized in the supplementary information.

This point was addressed to the best we can without compromising our intentions for this research. 

Because IgMs are very complex secretory glycoproteins, we wanted to ensure we first established a suitable experimental platform so that we could start asking questions on IgM biosynthesis to uncover something new. It was a natural process to go through many of the known important concepts in our own assay and protein expression systems. 

We judged it was too presumptuous to assume that a series of concepts proposed during the past 30 years (by using different clones and sources of IgMs from different species, different assays, and different cell types, etc.) are automatically recapitulated in our natural human IgM—partly because we don’t use the same assays that investigators in the 80s and 90s had used. We really needed to see how things could translate if we used our assay methods in our cell hosts. This is pointed out by Reviewer #2 in the comment, “as the differences between experimental systems may reveal novel phenomena”. Nonetheless, I tried to address this the best I could. 

The cell line we used, HEK293(6E), is widely used in the biotech industry as a preferred host to produce recombinant immunoglobulins during early drug discovery. It would be beneficial for the biotech community to fully demonstrate the potential of this cell host in making a complex molecule like IgM and how the cells respond to the demand of IgM assembly. Plus, our expression platform allowed us to switch the production of hexamers and pentamers qualitatively, which helped us to test the effects of mutations in hexamer and pentamer in a more controlled fashion. 

2. The text needs to be shortened. Often experiments and results are described in too much detail and length. For example, one could state that μHC expressed alone always formed Russel bodies regardless of mutations – with the exception of 4x C>S. A description of each mutant in this context is overwhelming if no additional effects are visible.

Agreed not to describe experiments and results in too much detail and length. 

3. 3.1 - The first paragraphs are meticulously performed proofs of principle. I suggest shifting these to the supplements, as they do not contain new insights into the topic of IgM biosynthesis and assembly. Figure 1 and 2 belong in the supplement section. The same is true for 3.2. Please highlight the one difference you spotted when co-expressing μ-HC-ΔCH1 with λLC. This information gets lost in the presence of the list of already known facts. The corresponding figures should be moved to the supplement. Please only show images for new findings.

The reviewer’s points are carefully considered. The other reviewer (Reviewer #2), however, requested us to show more characterization of IgM puncta by co-staining with the ER exit site or ERGIC markers to identify the structure. Likewise, Reviewer #2 asked us to co-stain the μHC induced Russell body with additional ER and Golgi markers. Furthermore, Reviewer #2 specifically asked us to determine the detergent solubility of intracellular μHC and its mutants. 

As such, instead of sending Fig. 1 and Fig. 2 to supplement, I decided to reorganize the figure panels by removing unnecessary images and gels, while adding new data that Reviewer #2 requested. I hope I hit a reasonable mid-point that addressed the feedback from both reviewers. 

4. The finding that Russel bodies are not formed by μ-HC-ΔCH1 C414/575S, which also is secretion competent, should be highlighted.

Followed the reviewer’s suggestion. I also added the detergent solubility data suggested by the other reviewer at the same time to highlight the rescue from Russell body, acquisition of detergent solubility, and restoration of secretion. 

5. Please refer to this paper: https://doi.org/10.15252/embj.2021108518 in the context of chapter 3.5

Although the reviewer suggested citing Giannone et al. (2022) 's paper in chapter 3.5, I found Chapter 3.4 is a more suitable place to cite it. This paper addresses why the C575S mutation decreases the efficiency of covalent polymeric IgM formation by blocking the formation of transitory non-native disulfide required for IgM polymerization. 

6. The titration of JC DNA and the correlation with the amount of pentamer is a very nice experiment and should get way more spotlight in the context of this paper.

By following this reviewer’s suggestion, I rewrote section 3.6. I also modified the Abstract and Intro to capture this. Thank you for the suggestions.

7. It is interesting, that you were able to secrete JC in the absence of μHC and λLC – something that was not reported before. This finding is worth more investigation.

I agree. The free JC secretion has not been widely known. We are in the midst of characterizing this enigmatic JC subunit behavior in more detail for a future manuscript. 

Incidentally, Kawasaki et al (2024) just published a paper in PNAS reporting the function of JC as a chemokine (see link below). There are many things we still do not know about JC subunit. 

https://doi.org/10.1073/pnas.2318995121

8. 3.7 - The presence of pentamers and hexamers has been already published.

Yes. The presence of pentamer and hexamer mixtures has been known for a long time. In fact, I cited relevant papers in this section to reflect that. However, it has never been shown as clearly as this experiment that we can qualitatively switch hexamer production to pentamer production by simply titrating transfecting the DNAs of each subunit. In many studies, the investigators do not regulate the production of hexamer or pentamers, and some investigators report JC-free IgM pentamers that we did not detect in our expression platform. We might have benefited from this robust and preferred recombinant expression system widely employed in the biotech industry. 

9. 3.8 - This section needs to be shortened and many data should be presented as supplementary information

I followed this reviewer’s suggestion as much as possible. The other reviewer suggested that I examine the detergent solubility of the JC subunit in the presence of other co-transfected subunit(s) in various combinations. As a result, it was challenging to make this section short, but I still tried. 

10. The new finding about JC as part of Russel bodies depending on the presence of other chains is interesting. When CH1 is deleted, the effects stay the same. The evidence could be moved to the supplement and the text shortened.

The co-aggregation of JC into Russell body (of μHC or μHC-ΔCH1) was dependent on the Cys-575 in the absence of an LC subunit. 

I tried to shorten the text and use the supplement as much as possible. 

11. 3.9 - It is already known, that C575 and C414 are necessary for correct pentamer assembly. C575 has the more important role. For example Hiramoto et. al. (DOI: 10.1126/sciadv.aau1199) already showed that for the C414S mutation, in addition to ‘loose’ pentamers, also tetramers can be formed. The cause, why the oligomers are not formed in your setting is probably the absence of C575.

We included analytical SEC data (directly performed on harvested culture media) for C414S, C575S, and C414/575S mutants, both in hexameric and pentameric IgM expression settings. 

This SEC data also showed some mutants that could not assemble covalent IgM polymer still assembled polymeric IgMs partly through non-covalent protein-protein intera

---

## [Decision Letter · Decision Letter 1]

7 May 2024

Understanding the biosynthesis of human IgM SAM-6 through a combinatorial expression of mutant subunits that affect product assembly and secretion.

PONE-D-23-28036R1

Dear Dr. Hasegawa,

We’re pleased to inform you that your manuscript has been judged scientifically suitable for publication and will be formally accepted for publication once it meets all outstanding technical requirements.

Kind regards,

Sebastian D. Fugmann, Ph.D.

Academic Editor

PLOS ONE

Additional Editor Comments (optional):

Reviewers' comments:

Reviewer's Responses to Questions

**Comments to the Author**

1. If the authors have adequately addressed your comments raised in a previous round of review and you feel that this manuscript is now acceptable for publication, you may indicate that here to bypass the “Comments to the Author” section, enter your conflict of interest statement in the “Confidential to Editor” section, and submit your "Accept" recommendation.

Reviewer #1: All comments have been addressed

Reviewer #2: (No Response)

2. Is the manuscript technically sound, and do the data support the conclusions?

Reviewer #1: Yes

Reviewer #2: Yes

3. Has the statistical analysis been performed appropriately and rigorously? 

Reviewer #1: Yes

Reviewer #2: N/A

4. Have the authors made all data underlying the findings in their manuscript fully available?

Reviewer #1: Yes

Reviewer #2: Yes

5. Is the manuscript presented in an intelligible fashion and written in standard English?

Reviewer #1: Yes

Reviewer #2: Yes

6. Review Comments to the Author

Reviewer #1: The authors made a great effort to answer my queries and the revised version is significantly improved.

The nature of the study is a systematic comparision of IgM mutant behaviour and this results in an extensive data set.

It is now more digestable and will be a valuable resource for the IgM field.

As well it contains intriuging new data.

Reviewer #2: This manuscript describes an impressive series of experiments aimed to dissect the mechanisms that promote the biogenesis of a complex secretory molecule, IgM. The experiments are carefully performed and technically solid. Despite most of them are confirmatory and essentially descriptive, these important results deserve to be made available to the scientific community.

Both reviewers suggested to show most confirmatory results as supplementary material, streamlining the paper qand focussing on the mechanistic aspects of the results described.

The authors clearly made an effort in this direction.

This reviewer would have been more drastic in pruning the main figures. However, this is essentially a question of taste that goes beyond the quality of the results.

So, apart from an encouragement to further simplify the presentation, this reviewer considers this paper suitable for publication.

7. PLOS authors have the option to publish the peer review history of their article (what does this mean?). If published, this will include your full peer review and any attached files.

Reviewer #1: No

Reviewer #2: No

---

## [Editor Report · Acceptance letter]

13 May 2024

PONE-D-23-28036R1 

PLOS ONE

Dear Dr. Hasegawa, 

I'm pleased to inform you that your manuscript has been deemed suitable for publication in PLOS ONE. Congratulations! Your manuscript is now being handed over to our production team.

Kind regards, 

on behalf of

Dr. Sebastian D. Fugmann 

Academic Editor

PLOS ONE